# Smoothed Analysis of Sequential Probability Assignment

**Alankrita Bhatt**
California Institute of Technology
abhatt@caltech.edu

**Nika Haghtalab**
University of California, Berkeley
nika@berkeley.edu

**Abhishek Shetty**
University of California, Berkeley
shetty@berkeley.edu

## Abstract

We initiate the study of smoothed analysis for the sequential probability assignment problem with contexts. We study information-theoretically optimal minmax rates as well as a framework for algorithmic reduction involving the *maximum likelihood estimator* oracle. Our approach establishes a general-purpose reduction from minimax rates for sequential probability assignment for smoothed adversaries to minimax rates for transductive learning. This leads to optimal (logarithmic) fast rates for parametric classes and classes with finite VC dimension. On the algorithmic front, we develop an algorithm that efficiently taps into the MLE oracle, for general classes of functions. We show that under general conditions this algorithmic approach yields sublinear regret.

## 1 Introduction

Sequential probability assignment — also known as online learning under the logarithmic loss — is a fundamental problem with far-reaching impact on information theory, statistics, finance, optimization, and sequential decision making [Rissanen, 1983, 1984, Cover, 1991, Feder et al., 1992, Xie and Barron, 1997, Merhav and Feder, 1998, Xie and Barron, 2000, Yang and Barron, 1999, Jiao et al., 2013, Orabona and Pál, 2016, Foster et al., 2018]. In recent years, methods for incorporating contexts or side information into sequential probability assignment have gained much attention [Rakhlin and Sridharan, 2015, Fogel and Feder, 2017, 2018, Foster et al., 2018, Bhatt and Kim, 2021, Bilodeau et al., 2021, Wu et al., 2022a], in part due to their newly forged connection to sequential decision making applications, the contextual bandit problem, and learning in Markov Decision Processes (MDPs) (see e.g. [Foster and Krishnamurthy, 2021] and [Foster et al., 2021a]). In this setting, a forecaster who has access to historical data $x_{1:t-1}, y_{1:t-1}$ consisting of contexts $x_\tau$ (e.g., day $\tau$'s meteorological information) and the outcomes $y_\tau \in \{0, 1\}$ (e.g., whether $\tau$ was a rainy day) wishes to predict $y_t$ given a new context $x_t$. The forecaster uses a *probability assignment* $p_t$ to estimate the probability of $y_t = 1$ outcome and incurs the logarithmic loss, i.e., $-\log p_t(y_t)$, which rewards the forecasters for having assigned high probability to the realized outcome. The goal of the forecaster is to suffer low *regret* against a chosen reference class of predictors.

A large body of prior work on sequential probability assignment with contexts has focused on settings where contexts are presented i.i.d. from an unknown distribution (see [Fogel and Feder, 2017, Bhatt and Kim, 2021, Bilodeau et al., 2021, Wu et al., 2022a] and the references within); this problem is also referred to as conditional density estimation. In these cases, sequential probability assignment is known to enjoy small regret for several reference classes such as Vapnik–Chervonenkis (VC) classes. On the other hand, attempts to consider context distributions that evolve unpredictably and

37th Conference on Neural Information Processing Systems (NeurIPS 2023).

adversarially have faced strong impossibility results even for simple reference classes of predictors. For example, for the reference class of simple one-dimensional thresholds assigning $p_t = \theta_0 \mathbb{1}\{x_t \leq a\} + \theta_1 \mathbb{1}\{x_t > a\}$ for $a \in [0, 1]$, regret is bounded by $O(\log T)$ in the i.i.d. case [Fogel and Feder, 2017] but is lower bounded by $\Omega(T)$ when the sequence of contexts is chosen adversarially (folklore e.g. Littlestone [1988]). In the face of the increasing need to adapt to evolving contexts in modern applications, these impossibility results indicate that new models of adversarial behavior must be considered for obtaining rigorous guarantees that guide the design of sequential probability assignment in practical applications.

In recent years, *smoothed analysis* of adaptive adversaries [Haghtalab et al., 2021, Rakhlin et al., 2011] has emerged as a framework for going beyond the worst-case adversaries while making minimal assumptions about the adaptive process that generates a sequence. In this setting, contexts are chosen from an evolving sequence of so-called $\sigma$-smooth distributions, whose density is bounded above by $1/\sigma$ times that of a base measure (such as the uniform distribution). Remarkably, these methods, established by Haghtalab et al. [2021] for 0-1 loss and extended to regression by Haghtalab et al. [2022], Block et al. [2022], have established performance guarantees for the sequential prediction problem that matches the optimal performance in the i.i.d setting. This raises the question as to whether the *sequential probability assignment* problem may similarly enjoy improved minmax regret bounds for smoothed adaptive sequences.

Beyond minmax rates, an important feature of probability assignment and, its analogue, density estimation is the availability of fundamental and natural estimation techniques such as *maximum likelihood estimation (MLE)*. For i.i.d. sequences, under general conditions, MLE is known to be optimal asymptotically and often serves as a starting point for designing more sophisticated estimators. Going beyond i.i.d. sequences, we ask whether MLE can be made to achieve good statistical behavior on adaptive sequences. More generally, algorithmic perspective is increasingly important for the sequential probability assignment problem and its applications to contextual bandits and reinforcement learning (where algorithm design is as fundamental a consideration as minmax rates [Agarwal et al., 2014, Simchi-Levi and Xu, 2022, Foster and Rakhlin, 2020, Langford et al., 2007, Foster et al., 2021b]). In this space, *oracle-efficient* sequential decision making algorithms that repurpose existing offline algorithmic routines have received special interest [Kalai and Vempala, 2005, Dudík et al., 2020, Wang et al., 2022, Kakade et al., 2007, Simchi-Levi and Xu, 2022]. Here again, recent progress on smoothed analysis for sequential prediction with 0-1 loss and regression loss [Haghtalab et al., 2021, Block et al., 2022] has shown promise in bridging the computational and information-theoretical gaps between what is obtainable in the i.i.d. case and for smoothed adaptive sequences.

In this paper, we initiate the study of smoothed analysis for sequential probability assignment and seek to understand fundamental information-theoretic limits on the *minmax regret* as well as design *natural and oracle-efficient algorithms* for this problem. Additionally, we investigate whether in the smoothed analysis setting, maximum likelihood estimation can efficiently address sequential probability assignment while achieving small regret. To the best of our knowledge, our work is the first to consider oracle-efficient algorithms (and particularly the MLE) for the sequential probability assignment problem.

## 1.1 Main results

**Reduction to transductive learning.** Our first main result is a reduction from regret bounds against a smoothed adversary to regret bounds against (a generalized version of) a transductive adversary. That is, we show that the minimax regret in the smoothed analysis setting is upper bounded by the minimax regret in the setting where a set of contexts is provided to the learner and the adversary is constrained to picking the contexts from this set. For $\mathcal{F}$, a class of hypotheses mapping contexts to $[0, 1]$, let us define the minmax regret in the transductive case over $T$ times steps when a context set of size $M$ is provided to the learner to be $\underline{\mathcal{R}}_T^M(\mathcal{F})$. We establish in Theorem 3.1 that for all $\sigma$-smooth sequences the minmax regret $\mathcal{R}_T(\mathcal{F}, \sigma)$ satisfies, for any $k > 1$,

$$\mathcal{R}_T(\mathcal{F}, \sigma) \leq \underline{\mathcal{R}}_T^{kT}(\mathcal{F}) + T^2(1 - \sigma)^k.$$

Furthermore, in Theorem 3.4 we upper bound $\underline{\mathcal{R}}_T^{kT}(\mathcal{F})$ by connecting the worst case adversarial regret in this setting to the scale-sensitive VC dimension of $\mathcal{F}$ which is a prototypical offline complexity of the class. Our results obtain a logarithmic dependence on $1/\sigma$. In particular, in

Corollary 3.4.1, we show that for VC classes (and parametric classes) the regret is bounded by $\mathcal{R}_T(\mathcal{F}, \sigma) \leq O\left(d \log\left(\frac{T}{\sigma}\right)\right)$, where $d$ is the VC dimension of class $\mathcal{F}$.

**Efficient Reduction from Sequential Probability Assignment to MLE.** Our second contribution is initiating the study of oracle-efficient algorithm design for sequential probability assignment. In particular, for small alphabet size, we design a natural algorithm (Algorithm 1) that efficiently uses an MLE oracle and achieves sublinear regret in the smoothed setting. Our Theorem 4.1 gives a general regret bound in terms of the statistical complexity of the class $\mathcal{F}$ and the smoothness parameter $\sigma$. For VC classes, this achieves regret rate of $T^{4/5}\sqrt{\frac{d}{\sigma}}$. To the best of our knowledge, this is the first *oracle-efficient* algorithm and analysis of the follow-the-perturbed-leader style algorithms for the logarithmic loss.

**Probability assignment for VC classes.** For VC classes $\mathcal{F}$, we explicitly construct sequential probability assignments and establish their regret guarantees in the smoothed setting. That is, we construct a probability assignment based on a Bayesian mixture over $\mathcal{F}$ that satisfies $\mathcal{R}_T(\mathcal{F}, \sigma) \leq Cd \log\left(\frac{T}{\sigma}\right)$ where $d$ is the VC dimension of class $\mathcal{F}$. While this approach is not oracle-efficient, it indeed achieves regret bound with optimal dependence on $T$ and $\sigma$. This motivates a natural direction for future work as to whether such mixture-based methods can be implemented oracle efficiently or if there is a tradeoff between the regret and the computational complexity of sequential probability assignment.

## 2 Preliminaries

Let $\mathcal{X}$ be a set of *contexts* and $\mathcal{Y} = \{0, 1\}$. Then, the problem being studied entails a sequential game where at each timestep $t$, based on the history of contexts $x_{1:t} := (x_1, \ldots, x_t)$ where $x_i \in \mathcal{X}$ and associated bits $y_{1:t-1} \in \{0, 1\}^{t-1}$, the player must assign a probability $q(\cdot | x_{1:t}, y_{1:t-1})$ to what the upcoming bit $y_t$ will be. Once the bit $y_t$ is revealed (possibly in an adversarial fashion) the player incurs loss $-\log q(y_t | x_{1:t}, y_{1:t-1})$ and the game proceeds to the next step. For a *hypothesis class* $\mathcal{F} \subset \{\mathcal{X} \to [0, 1]\}$, the associated regret for a probability assignment strategy $\mathscr{Q} = \{q(\cdot | x_{1:t}, y_{1:t-1})\}_{t=1}^n$ for a fixed $x_{1:T}, y_{1:T}$ is

$$\mathcal{R}_T(\mathcal{F}, x_{1:T}, y_{1:T}, \mathscr{Q}) = \sum_{t=1}^T \log \frac{1}{q(y_t | x_{1:t}, y_{1:t-1})} - \inf_{f \in \mathcal{F}} \sum_{t=1}^T \log \frac{1}{p_f(y_t | x_t)} \tag{1}$$

where $p_f(1 | x_t) = f(x_t)$; i.e. the function $f$ assigns probability $\mathrm{Bern}(f(x_t))$ to the upcoming bit given the context $x_t$.

Our statistical results apply to a general loss function $\ell$ and general actions of the learner $a_t$. For a set of inputs $\{(x_i, y_i)\}_{i=1}^T$ specified by the adversary and a set of actions $\{a_i\}_{i=1}^T$ of the learner, regret is defined by

$$\mathcal{R}_T(\mathcal{F}, x_{1:T}, y_{1:T}, a_{1:T}) = \sum_{t=1}^T \ell(a_t, (x_t, y_t)) - \inf_{f \in \mathcal{F}} \sum_{t=1}^T \ell(f(x_t), (x_t, y_t)),$$

where for log-loss $a_t = q(\cdot | x_{1:t}, y_{1:t-1})$; the action is a probability mass function (pmf) over $\{0, 1\}$.

The regret in (1) is often studied under various adversary models; i.e. various different probabilistic assumptions (or lack thereof) on the model generating $x_t$ and $y_t$. In this work, we consider worst-case $y_t$ (in contrast to the *realizable* setting where $Y_t \sim \mathrm{Bern}(f^*(x_t))$ for a fixed unknown $f^* \in \mathcal{F}$) and $X_t \sim \mathcal{D}_t$ where $\mathcal{D}_t$s form an adaptive sequence of smooth distributions.

**Definition 2.1** (Smooth distribution and adversary [Haghtalab et al., 2020]). Consider a fixed and known base distribution $\mu$ on $\mathcal{X}$ (such as the uniform distribution if $\mathcal{X}$ supports it). A distribution $\mathcal{D}$ on $\mathcal{X}$ is said to be $\sigma$-smooth if for all measurable sets $A \subseteq \mathcal{X}, \mathcal{D}(A) \leq \frac{\mu(A)}{\sigma}$. We denote the set of all $\sigma$-smooth distributions by $\Delta_\sigma(\mu)$. An adversary, characterized by a joint distribution $\mathscr{D}$ with $X_t \sim \mathcal{D}_t$ (where $\mathcal{D}_t$ may possibly depend on the history) is said to be a $\sigma$-smooth adaptive adversary if $\mathcal{D}_t \in \Delta_\sigma(\mu)$ for all $t \in \{1, \ldots, T\}$.

The minmax regret for $\sigma$-smooth adaptive adversaries is then given by

$$\mathcal{R}_T(\mathcal{F}, \sigma) = \inf_{\mathscr{Q}} \sup_{\sigma\text{-smoothed } \mathscr{D}} \mathbb{E}_{X_{1:T} \sim \mathscr{D}} \left[ \max_{y_{1:T}} \mathcal{R}_T(\mathcal{F}, X_{1:T}, y_{1:T}, \mathscr{Q}) \right],$$

where $\mathscr{Q}$ is the set of all probability assignment strategies.

We are particularly interested in how geometric properties of the function class $\mathcal{F}$ affect $\mathcal{R}_T(\mathcal{F}, \sigma)$. There are several notions of covering numbers and combinatorial dimensions that quantify the "richness" and complexity of a class, but the *scale-sensitive VC dimension* will be of particular interest to us and is invoked in our results.

**Definition 2.2** (Scale-sensitive VC dimension). Let $\mathcal{F}$ be a function class. For any $\alpha > 0$ and points $x_1, \ldots, x_m \in \mathcal{X}$, we say that $\mathcal{F}$ shatters the set $x_1, \ldots, x_m$ at scale $\alpha$ if there exist $s_1 \ldots s_m \in \mathbb{R}$ such that for each $\epsilon \in \{-1, 1\}^n$ there exists a function $f \in \mathcal{F}$ such that $\epsilon_i \left( f(x_i) - s_i \right) \geq \frac{\alpha}{2}$. The scale sensitive VC dimension at scale $\alpha$ of $\mathcal{F}$, denoted by $\mathrm{VC}(\mathcal{F}, \alpha)$ is defined as the largest $m$ such that there is a set of $m$ points $x_1, \ldots, x_m \in \mathcal{X}$ such that $\mathcal{F}$ shatters the set at scale $\alpha$. The (traditional) VC dimension of a binary class $\mathcal{F}$ is defined as $\mathrm{VC}(\mathcal{F}) = \lim_{\alpha \to 0^+} \mathrm{VC}(\mathcal{F}, \alpha)$.

Throughout, we use the following result of Haghtalab et al. [2021] about $\sigma$-smooth distributions. This results aids us in reduction from smoothed learning to transductive learning.

**Theorem 2.1** (Coupling Lemma of Haghtalab et al. [2021]). *Let $\mathscr{D}_\sigma$ be an adaptive sequence of $t$ $\sigma$-smooth distributions on $\mathcal{X}$. There is a coupling $\Pi$ such that*

$$(X_1, Z_{1,1}, \ldots, Z_{1,K}, \ldots, X_t, Z_{t,1}, \ldots, Z_{t,K}) \sim \Pi$$

1. *$X_1, \ldots, X_t$ is distributed according $\mathscr{D}_\sigma$,*

2. *For every $j \leq t$, $\{Z_{i,k}\}_{i \geq j, k \in [K]}$ are uniformly and independently distributed on $\mathcal{X}$, conditioned on $X_1, \ldots, X_{j-1}$.*

3. *For any $t$, with probability at least $1 - (1 - \sigma)^K$, $X_t \in \{Z_{t,k}\}_{k=1:K}$.*

## 3 General reduction to transductive learning

In this section, we will consider the minimax regret for the smoothed online learning game with respect to the loss function[1] $\ell$ against a general class of functions $\mathcal{F}$. In Section 3.1, we will show that the minimax regret can be reduced to the minimax regret for a version of transductive learning with respect to the same loss function and class of functions. In Section 3.2, we give general upper bounds for the transductive setting. There is a subtle but important difference between reduction that directly involve regret compared to recent efforts (such as Haghtalab et al. [2021], Block et al. [2022], Haghtalab et al. [2022]) using reductions between proxies of regret, such as covering numbers and sequential complexities. This is particularly important for log loss since its complexity is not captured by covering numbers. We discuss this point further in Section 3.2.2.

### 3.1 Regret-to-regret Reduction

We work with a general loss function $\ell$ and general actions of the learner $a_t$. We note that, we can write the minimax value of the smoothed setting in extensive form as

$$\mathcal{R}_T(\mathcal{F}, \sigma) = \sup_{\mathcal{D}_1 \in \Delta_\sigma(\mu)} \mathbb{E}_{X_1 \sim \mathcal{D}_1} \inf_{a_1} \sup_{y_1} \sup_{\mathcal{D}_2 \in \Delta_\sigma(\mu)} \mathbb{E}_{X_2 \sim \mathcal{D}_2} \inf_{a_2} \sup_{y_2} \ldots \tag{2}$$

$$\ldots \sup_{\mathcal{D}_T \in \Delta_\sigma(\mu)} \mathbb{E}_{X_T \sim \mathcal{D}_T} \inf_{a_T} \sup_{y_T} \mathcal{R}(\mathcal{F}, X_{1:T}, y_{1:T}, a_{1:T}).$$

In order to bound this, we consider a generalization of the notion of online learning that is referred to as transductive learning. In this setting, at the start of the interaction the adversary chooses a set of contexts $X = \{X_i\}_{i=1}^M$ for some $M \geq T$ and provides this to the player. The game proceeds as

---

[1]The reduction works for general loss functions with the property that the worst-case regret for horizon $T$ is bounded by $T$, but one can think about $\ell$ being the log-loss throughout this section for concreteness.

before with the adversary picking $(x_t, y_t)$ at time $t$ and the learner picking an action $a_t$ and suffering a loss $\ell(a_t, (x_t, y_t))$. However, the adversary is now constrained to pick $x_t \in X$ at all times $t$. We can then define the minmax regret indexed by $X$ as

$$\underline{\mathcal{R}}_T(\mathcal{F}, X) := \left[ \max_{x_1 \in X} \inf_{a_1} \sup_{y_1} \dots \max_{x_T \in X} \inf_{a_T} \sup_{y_T} \mathcal{R}(\mathcal{F}, x_{1:T}, y_{1:T}, a_{1:T}) \right].$$

Furthermore, define the worst-case transductive learning regret for sets of size $M$ as $\underline{\mathcal{R}}_T^M(\mathcal{F}) = \max_{X \subseteq \mathcal{X}, |X| = M} \underline{\mathcal{R}}_T(\mathcal{F}, X)$. In the following theorem, we show that the regret against $\sigma$-smoothed adversaries is bounded by the regret in the transductive learning setting when the set of contexts is drawn from the base distribution $\mu$.

**Theorem 3.1.** *Let $\mathcal{F}$ be any class of functions from $\mathcal{X}$ to $\mathbb{R}$ and let $\sigma \in (0, 1]$. Then, for any $T$ and $k$, we have*

$$\mathcal{R}_T(\mathcal{F}, \sigma) \leq \underline{\mathcal{R}}_T^{kT}(\mathcal{F}) + T^2(1 - \sigma)^k.$$

Theorem 3.1 shows that we can reduce the problem of evaluating the minimax regret for smoothed adversaries to evaluating the minimax regret for transductive learning. Note that the second term $(1 - \sigma)^k \leq e^{-k\sigma}$ and thus, in order to get bounds that are sublinear one needs to consider $k = c \log T / \sigma$ for an appropriate absolute constant $c$. As we will see in the next section, this leads to logarithmic dependence on $\sigma^{-1}$. Moreover, by Theorem 3.1 and since $\mathbb{E}_{X \sim \mu^T} \underline{\mathcal{R}}_T(\mathcal{F}, X) \leq \mathcal{R}_T(\mathcal{F}, \sigma)$, we can see that the transductive learning regret exactly captures the smoothed regret up to $\mathrm{polylog}\left(\frac{T}{\sigma}\right)$ factors.

### 3.2 Bounds for Transductive Learning

In this section, we discuss ways to upper bound transductive learning regret $\underline{\mathcal{R}}_T^M(\mathcal{F})$ so as to achieve bounds on $\mathcal{R}_T(\mathcal{F}, \sigma)$ via Theorem 3.1.

#### 3.2.1 Using Covering Numbers

One of the approaches common in online learning is to characterize the regret in terms of geometric properties (such as covering numbers) of the function class $\mathcal{F}$. The notion of covering required varies depending on the loss function and the stochastic properties of the data—typically completely adversarial problems require stronger notions of sequential coverings [Rakhlin et al., 2015a,b] while for stochastic problems usually weaker *offline coverings* suffice. In our smoothed case, we show that the offline complexity notion of scale-sensitive VC dimension as defined in Definition 2.2 is adequate. Similar ideas were considered for the case of regression and convex Lipshitz losses in Haghtalab et al. [2022], Block et al. [2022].

Let us first define the notion of approximation according to which a cover will be constructed; we will consider a pointwise approximation. This notion is similar to the notion of global sequential covering in Wu et al. [2022b].

**Definition 3.1.** *Let $\mathcal{F}$ be a function class. A set of functions $\tilde{\mathcal{F}}$ is said to be a $\epsilon$-covering of $\mathcal{F}$ if for any $f \in \mathcal{F}$ there exists $g \in \tilde{\mathcal{F}}$ such that $\sup_{x \in \mathcal{X}} |f(x) - g(x)| \leq \epsilon$. We will use $\mathcal{N}(\mathcal{F}, \epsilon)$ to denote the size of the minimal $\epsilon$-covering of $\mathcal{F}$.*

Note that while the metric in Definition 3.1 is quite stringent, using this cover in the transductive learning case requires us to only consider function classes with *bounded domain size*. We capture this using the following theorem.

**Theorem 3.2** (Upper bound on transductive learning)**.** *Let $\mathcal{F}$ be a function class and $\epsilon > 0$. Then,*

$$\underline{\mathcal{R}}_T^{kT}(\mathcal{F}) \leq \inf_{\epsilon} \left\{ \sup_{Z \subset \mathcal{X}, |Z| = kT} \log \mathcal{N}\left(\mathcal{F}|_Z, \epsilon\right) + 2\epsilon T \right\},$$

*where $\mathcal{F}|_Z$ is the projection of hypothesis class $\mathcal{F}$ on the set $Z$.*

The proof of this theorem follows from relating transductive learning to the worst sequential prediction on a finite set of points using formalism presented in Wu et al. [2022b]. The proof of this theorem is deferred to the Appendix C.

Next, we recall that the covering number $\mathcal{N}(\mathcal{F}, \epsilon)$ is bounded as a function of the scale sensitive VC dimension of the class $\mathcal{F}$ and the number of points in the domain.

**Theorem 3.3** (Rudelson and Vershynin [2006])**.** *There exist universal constants $c, C$ such that for all $\alpha > 0$, any function class $\mathcal{F}$ defined on a finite set $\mathcal{X}$, and $\epsilon > 0$, we have*

$$\log \mathcal{N}(\mathcal{F}, \epsilon) \leq C \cdot \text{VC}(\mathcal{F}, c\alpha\epsilon) \log^{1+\alpha} \left( \frac{C|\mathcal{X}|}{\text{VC}(\mathcal{F}, c\epsilon)\epsilon} \right).$$

Finally, putting together Theorem 3.1, Theorem 3.2 and Theorem 3.3 we get the following.

**Theorem 3.4** (Minimax smoothed regret and scale-sensitive VC dimension)**.**

$$\mathcal{R}_T(\mathcal{F}, \sigma) \leq \inf_{k, \alpha, \epsilon > 0} \left\{ C \cdot \text{VC}(\mathcal{F}, c\alpha\epsilon) \log^{1+\alpha} \left( \frac{CkT}{\text{VC}(\mathcal{F}, c\epsilon)\epsilon} \right) + 2\epsilon T + T^2 (1-\sigma)^k \right\}.$$

We can instantiate the bound in Theorem 3.4 in terms of $T$ and $\sigma$ for two particularly interesting cases: when $\text{VC}(\mathcal{F}, \epsilon)$ scales as $d \log (1/\epsilon)$ (often referred to as parametric classes) and when $\text{VC}(\mathcal{F}, \epsilon)$ scales as $\epsilon^{-p}$ (often referred to as nonparametric classes). A canonical example of the former are *VC classes*; for a class with VC dimension $d$, $\text{VC}(\mathcal{F}^{\text{VC}}, \epsilon) = Cd \log \left( \frac{1}{\epsilon} \right)$ (see for example [Vershynin, 2018, Theorem 8.3.18]). A canonical example of the latter are functions of bounded variation, $\mathcal{F}^{\text{BV}}$ which have $\text{VC}(\mathcal{F}^{\text{BV}}, \epsilon) = \frac{C}{\epsilon}$ (see for example [Musayeva, 2020, Bartlett et al., 1997]). This class is known to have unbounded sequential covering numbers [Rakhlin et al., 2010] and therefore is not learnable with a worst-case adversary—this can be seen as a simple consequence of the fact that $\mathcal{F}^{\text{BV}}$ contains all one-dimensional thresholds.

**Corollary 3.4.1** (Rates for parametric and nonparametric classes)**.** *If $\text{VC}(\mathcal{F}, \epsilon) = d \log (1/\epsilon)$, then for a large enough $T$*

$$\mathcal{R}_T(\mathcal{F}, \sigma) \leq O \left( d \cdot \text{poly} \log \left( \frac{T}{\sigma} \right) \right).$$

*If $\text{VC}(\mathcal{F}, \epsilon) = \epsilon^{-p}$, then*

$$\mathcal{R}_T(\mathcal{F}, \sigma) \leq O \left( T^{\frac{p}{p+1}} \cdot \text{poly} \log \left( \frac{T}{\sigma} \right) \right).$$

In particular, note that Corollary 3.4.1 shows that for VC classes $\mathcal{R}_T(\mathcal{F}^{\text{VC}}, \sigma) = \widetilde{O} \left( d \cdot \log \left( \frac{T}{\sigma} \right) \right)$ (tight, see also concurrent work [Wu et al., 2023] for a similar bound) and for functions of bounded variation $\mathcal{R}_T(\mathcal{F}^{\text{BV}}, \sigma) = \widetilde{O} \left( \sqrt{T} \log \left( \frac{1}{\sigma} \right) \right)$; note that the minmax rates for the worst-case adversary scale as $\Omega(T)$ for both these cases. We note that the above bound may be loose for general nonparametric classes, but should be improvable using a multiscale (chaining) version of Theorem 3.4 but we do not focus on this here.

Though the above results give satisfactory bounds in the minimax sense for many classes $\mathcal{F}$ of interest, it is useful to consider explicit constructions of probability assignment rules. For the case of finite VC dimension, we give an explicit probability assignment rule by considering a discretization of the class and using a mixture probability assignment rule. In particular, this strategy (denoted by $\mathcal{Q}^{\text{VC}}$) yields optimal regret $\mathcal{R}_T(\mathcal{F}^{\text{VC}}, \sigma, \mathcal{Q}^{\text{VC}}) \leq O \left( d \log \left( \frac{T}{\sigma} \right) \right)$. For the formal statements, proofs and detailed discussion see Appendix D.

### 3.2.2 Examples without Covering numbers

This reduction approach to characterizing minmax regret in the log-loss is interesting since all previous approaches have used covering numbers of some kind—either sequential covering numbers

or stronger notions of global covering. However, in stark contrast to the 0/1 loss and several other loss functions [Rakhlin et al., 2015b], covering numbers cannot capture the minmax regret for the log-loss, at least in the adversarial case. Consider the following class of functions on context set $\mathcal{X} = \mathbb{B}_2$ (where $\mathbb{B}_2$ denotes the unit $\ell_2$ Euclidean ball) $\mathcal{F}^{\text{Lin}} := \left\{ x \mapsto \frac{\langle x, w \rangle + 1}{2} \,\middle|\, w \in \mathbb{B}_2 \right\}$. For this class, Rakhlin and Sridharan [2015] construct a follow-the-regularized leader (FTRL) based algorithm achieving regret $O(\sqrt{T})$. However, Bilodeau et al. [2020] show an upper bound on the regret in terms of sequential covering numbers which is not improvable in general—this shows that sequential covering numbers are not adequate to capture the minmax regret rates for the log-loss. Wu et al. [2022b] further consolidate this by considering the following class, closely resembling $\mathcal{F}^{\text{Lin}}$, $\mathcal{F}^{\text{AbsLin}} := \left\{ x \mapsto |\langle x, w \rangle| \,\middle|\, w \in \mathbb{B}_2 \right\}$. [Wu et al., 2022b, Example 2, Theorem 6] establish that the minmax regret for $\mathcal{F}^{\text{AbsLin}}$ is $\widetilde{\Theta}(T^{2/3})$, demonstrating the surprising fact that by a simple linear transformation of the hypothesis class (which does not change its covering number) one can obtain minmax rates that differ by a polynomial factor! On the other hand, our reduction-based approach bypasses the need for using any covering based arguments and therefore would lead to tight (at least up to poly $\log(T/\sigma)$) rates.

We remark that exact characterizations of the minmax regret with log-loss (often referred to as the minmax redundancy in the information theory literature) in the no-context (adversarial) case is most often calculated by studying the so-called *stochastic complexity* of the class $\mathcal{F}$ [Rissanen, 1996]. This can be extended to (worst-case) transductive learning with contexts $x_1, \ldots, x_T$; in this case the minmax optimal regret for a fixed horizon is achieved by the normalized maximum likelihood (NML) probability assignment [Shtar'kov, 1987], and can be expressed as $\underline{\mathcal{R}}_T^T(\mathcal{F}) = \max_{x_{1:T}} \log \left( \sum_{y_{1:T} \in \{0,1\}^T} \max_{f \in \mathcal{F}} \prod_{t=1}^T p_f(y_t | x_t) \right)$. This expression has been evaluated previously for online logistic regression [Jacquet et al., 2021] and more general hypothesis classes [Wu et al., 2022b]. It is an intriguing question to understand what properties of $\mathcal{F}$ the stochastic complexity depends on, given that the above examples illustrates that covering numbers do not capture it. Our reduction provides a technique to use such fine-grained understanding of the regret to directly lift the bounds to the more general smoothed adversary setting.

# 4 Oracle-Efficient Smoothed Sequential Probability Assignment

In the previous section, we consider a purely statistical perspective on the minmax value of the sequential probability assignment problem for smoothed adversaries. In this section, we will focus on an algorithmic perspective and design an algorithm that is efficiently implemented using calls to an MLE oracle. We will focus on the setting when the base measure is the uniform measure on the input space $\mathcal{X}$ and the label space $\mathcal{Y} = \{0, 1\}$. In this setting, we are given access to an oracle OPT which given a data set $S = \{x_i, y_i\}_{i=1}^m$ outputs a hypothesis that minimizes the loss on $S$. That is,

$$\text{OPT}\,(S) = \operatorname*{argmin}_{h \in \mathcal{F}} \frac{1}{m} \sum_{i=1}^m \ell\left(h, (x_i, y_i)\right).$$

In the context of the logarithmic loss, this corresponds to maximum likelihood estimation. Most of the analysis holds for a general loss function $\ell$ (with regret scaling appropriate bounds on the values and derivatives), but for clarity one can think of $\ell$ as the log-loss. In particular, Algorithm 1 is written for the log-loss.

The main framework we work in is the follow-the-perturbed-leader (FTPL) framework. Here, our algorithm uses the oracle on a data set consisting of the historical samples and a set of hallucinated samples. The hallucinated samples are intended to "stabilize" the predictions of the algorithm. This gives us a probability assignment $\mathscr{Q}^{\text{FTPL}} = \{q^{\text{FTPL}}(\cdot | x_{1:t}, y_{1:t-1})\}_{t=1}^T$.

In order to state the regret bound, we need the following notions. For any class $\mathcal{F}$, define the truncated class $\mathcal{F}_\alpha$ as $\mathcal{F}_\alpha = \left\{ f_\alpha : f_\alpha(x) = \frac{f(x) + \alpha}{1 + 2\alpha} \text{ where } f \in \mathcal{F} \right\}$. We will also need the notion of Rademacher complexity $\text{Rad}\,(\mathcal{F}, T) = \sup_{X \subset \mathcal{X}, |X| = T} \mathbb{E}_\epsilon \left[ \sup_{f \in \mathcal{F}} \frac{1}{T} \sum_{x \in X} \epsilon_x f(x) \right]$.

**Algorithm 1:** Probability assignment $\mathscr{Q}^{\text{FTPL}}$

---

**Input:** time horizon $T$, smoothness parameter $\sigma$, VC dimension $d$, Samples per step Parameter $n$, Truncation parameter $\alpha$

**1 for** $t \leftarrow 1$ **to** $T$ **do**

**2**     Generate $N^{(t)} \sim \text{Poi}(n)$ fresh hallucinated samples $(\widetilde{x}_1^{(t)}, \widetilde{y}_1^{(t)}), \cdots, (\widetilde{x}_N^{(t)}, \widetilde{y}_N^{(t)})$, which are i.i.d. conditioned on $N$ with $\widetilde{x}_i^{(t)} \sim \mathcal{U}(\mathcal{X})$ and $\widetilde{y}_i^{(t)} \sim \mathcal{U}(\{0,1\})$ Call the oracle to compute $\tilde{h}_t \leftarrow \text{OPT}\left(\{(\widetilde{x}_i^{(t)}, \widetilde{y}_i^{(t)})\}_{i \in [N^{(t)}]} \cup \{x_\tau, y_\tau\}_{\tau \in [t-1]}\right)$

**3**     Observe $x_t$

**4**     Assign probability $q^{\text{FTPL}}(1|x_{1:t}, y_{1:t-1}) = \frac{\tilde{h}_t(x_t) + \alpha}{1 + 2\alpha}$

**5**     Receive $y_t$

**6 end**

---

**Theorem 4.1** (Main Regret Bound). *For any hypothesis class $\mathcal{F}$ and parameters $n, \alpha$, we have that the regret of Algorithm 1 for $\sigma$-smoothed adversaries is bounded as*

$$\mathcal{R}_T(\mathcal{F}, \sigma, \mathscr{Q}^{\text{FTPL}}) \leq n \log\left(\frac{1}{\alpha}\right) + \alpha T + T\sqrt{\log\left(\frac{1}{\alpha}\right) \cdot \frac{1}{\sigma n}}$$

$$+ T \cdot \inf_{m \leq n} \left\{\frac{1}{\alpha}\text{Rad}\left(\mathcal{F}_\alpha, n/m\right) + \frac{n(1-\sigma)^m \log\left(1/\alpha\right)}{m} + e^{-n/8}\right\}.$$

We will instantiate this bound for case when the class $\mathcal{F}$ has bounded VC dimension. For such classes, it is known that the Rademacher complexity is bounded. We state this in the following corollary.

**Corollary 4.1.1.** *Let $\mathcal{F}$ be a hypothesis class such that the Rademacher complexity is bounded as* $\text{Rad}\left(\mathcal{F}_\alpha, T\right) = cT^{-\omega}$, *then we have* $\mathcal{R}_T(\mathcal{F}, \sigma, \mathscr{Q}^{\text{FTPL}}) \leq T^{\frac{2}{2+\omega}}\sqrt{\frac{1}{\sigma}} \cdot \text{poly} \log\left(\frac{Td}{\sigma}\right)$.

Note that, in particular, for VC classes we have $\mathcal{R}_T\left(\mathcal{F}^{\text{VC}}, \sigma\right)$ scales as $T^{4/5}$. Improving this to achieve the minimax rate discussed in Corollary 3.4.1 is an interesting open question.

*Remark.* A slightly improved regret scaling as $T^{3/4}$ can be achieved by assuming access to an oracle that can optimize a mixed objective function involving the log-loss and signed sum of the functions in class. However, these oracles do not have the natural interpretation in terms of maximum likliehood estimation.

## 4.1 Analysis

The main challenge in designing algorithms in the follow-the-perturbed-leader framework is designing the distribution of the hallucinated samples so as to balance the tradeoff between the "stability" of the algorithm i.e. how little the algorithm changes its prediction from time step to time step, and the "perturbation" i.e. how much the addition of the hallucinated samples changes the prediction of the algorithm from the outputting the best hypothesis on the historical samples. This is captured by the following lemma.

**Lemma 4.2** (Follow the Perturbed Leader bound). *Let $\ell$ be a convex loss function, and let $\mathcal{F}$ be a hypothesis class. Let $\mathcal{D}_t$ denote the distribution of the adversary at time $t$ and let $\mathcal{Q}_t$ denote the distribution of the hypothesis $h_t$ output by Algorithm 1. Then, we have that the regret of Algorithm 1 (where we use $\widetilde{s}_t = (\widetilde{x}_t, \widetilde{y}_t)$ to denote a hallucinated data point) [2] is bounded by*

$$\sum_{i=1}^T \underbrace{\mathbb{E}_{s_t \sim \mathcal{D}_t}\left(\mathbb{E}_{h_t \sim \mathcal{Q}_t}[\ell(h_t, s_t)] - \mathbb{E}_{h_{t+1} \sim \mathcal{Q}_{t+1}}[\ell(h_{t+1}, s_t)]\right)}_{\text{Stability}} + \underbrace{\mathbb{E}\left[\sup_{h \in \mathcal{F}_\alpha} \sum_{i=1}^N \ell(h, \tilde{s}_t) - \ell(h^*, \tilde{s}_t)\right]}_{\text{Perturbation}} + \alpha T.$$

*where $h^* = \text{argmin}_{h \in \mathcal{F}_\alpha} \sum_{i=1}^T \ell(h, s_t)$.*

---

[2]With some abuse of notation, we consider $\mathcal{D}_t$ to be over $\mathcal{X} \times \{0,1\}$.

We provide a proof in Appendix E for completeness. Given this decomposition of the regret, we need to handle both terms carefully. Just to appreciate the tradeoff, note that as we increase the number of hallucinated examples, the perturbation term generally increases, but the stability term generally decreases. First, let us focus on the stability term which is harder to deal with. The main approach we will use is a generalization of the decomposition of the stability term introduced in Haghtalab et al. [2022] even when the losses are unbounded, as is the case with the log-loss. The main idea is to decompose the stability term in terms of the distance between the distribution of the average prediction at the next time step and the distribution of the current time step, as captured by the $\chi^2$ distance and a term that captures how different the predictions of the algorithm are when the sample is resampled from the same distribution. The proof can be found in Appendix F. [3]

**Lemma 4.3** ( $\chi^2$ + Generalization $\Rightarrow$ Stability). *Let $\mathcal{Q}_t$ denote learner's distribution over $\mathcal{F}$ in at round $t$, $\mathcal{D}_t$ be adversary's distribution at time $t$ (given the history $s_1, \cdots, s_{t-1}$), $s_t \sim \mathcal{D}_t$ be the realized adversarial instance at time $t$, and $s'_t$ be an independent copy $s'_t \sim \mathcal{D}_t$. Let $R^{(t+1)}$ refers to the randomness used by the algorithm in round $t+1$. Then,*

$$
\mathop{\mathbb{E}}_{s_t \sim \mathcal{D}_t} \left( \mathop{\mathbb{E}}_{h_t \sim \mathcal{Q}_t} [\ell(h_t, s_t)] - \mathop{\mathbb{E}}_{h_{t+1} \sim \mathcal{Q}_{t+1}} [\ell(h_{t+1}, s_t)] \right)
$$
$$
\leq \sqrt{\frac{1}{2} \chi^2 (\mathop{\mathbb{E}}_{s_t \sim \mathcal{D}_t} [\mathcal{Q}_{t+1}], \mathcal{Q}_t) \cdot \log\left(\frac{1}{\alpha}\right)} + \mathop{\mathbb{E}}_{s_t, s'_t \sim \mathcal{D}_t; R^{(t+1)}} [\ell(h_{t+1}, s'_t) - \ell(h_{t+1}, s_t)].
$$

Given this lemma, we move on to bounding the $\chi^2$ divergence between the distribution of the average prediction at the next time step and the distribution of the current time step. This is done using the Ingster method to bound the divergence of mixtures. We include a proof in Appendix G for completeness.

**Lemma 4.4** (Bound on $\chi^2$). $\chi^2 \left( \mathbb{E}_{s_t \sim \mathcal{D}_t}[\mathcal{Q}_{t+1}], \mathcal{Q}_t \right) \leq \frac{2}{\sigma n}$.

Next, we move on second term in Lemma 4.3. Note that this term involves the difference between the loss of the hypothesis output at time $t+1$ evaluated on two independent points $s_t$ and $s'_t$ drawn from $\mathcal{D}_t$. The main idea to bound the term is to use a stronger version of the coupling lemma Theorem 2.1 which allows us to extract subsequences of points sampled according to smooth distributions from iid samples from the base measure. This allows us to relate the required generalization bound to the Rademacher complexity of the class composed with the loss. Using the truncation and the contraction principle, we get the desired bound. The proof can be found in Appendix H.

**Lemma 4.5** (Generalization). *Let $h_{t+1}$ denote the hypothesis output by the Algorithm 1 at time $t+1$. Then, for any $m \leq n$, we have*

$$
\mathop{\mathbb{E}}_{s_t, s'_t \sim \mathcal{D}_t; R^{(t+1)}} \left[ \ell(h_{t+1}, s'_t) - \ell(h_{t+1}, s_t) \right] \leq \frac{1}{\alpha} \mathrm{RAD} \left( \mathcal{F}_\alpha, n/m \right) + \frac{n \left(1 - \sigma\right)^m \log\left(1/\alpha\right)}{m} + e^{-n/8}.
$$

The final term that we want to bound is the perturbation term. In order to bound this note that we set our truncation parameter $\alpha$ such that the loss our the prediction made by our algorithm is bounded and consequently the perturbation term is bounded by $n \log \frac{1}{\alpha}$, see Appendix I. Theorem 4.1 follows by combining the above results.

## 5  Conclusions and Open Problems

In this paper, we initiated the study of sequential probability assignment with smoothed adversaries. We characterize the minimax regret in terms of the minimax regret for transductive learning and use this to provide tight regret bounds, e.g., for VC classes. Furthermore, we initiate the study of oracle efficiency in this setting and show that sublinear regret can be achieve for general classes.

Our work motivates several directions for future work. An interesting direction is whether the optimal $O(\log(T))$ regret is achievable for some classes, such as VC classes, using oracle-efficient algorithms. More generally, are there computational barriers to obtaining fast rates in prediction with log-loss in this setting?

---

[3]For the particular use in our analysis a simpler version of the lemma similar to Haghtalab et al. [2022] suffices but we prove a general version since we believe such a version is useful in providing improved regret bounds for the problem.

## Acknowledgments and Disclosure of Funding

This work was supported in part by the National Science Foundation under grant CCF-2145898, a C3.AI Digital Transformation Institute grant, and Berkeley AI Research Commons grants. This work was partially done while authors were visitors at the Simons Institute for the Theory of Computing.

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

# A    Related work

Sequential probability assignment is a classic topic in information theory with extensive literature, see the survey by Merhav and Feder [1998] and the references within. In particular, the idea of probability assignments that are Bayesian mixtures over the reference class of distributions [Krichevsky and Trofimov, 1981] is of central importance—such mixture probability assignments arise as the optimal solution to several operational information theoretic and statistical problems [Kamath et al., 2015]. It is also known that the Bayesian mixture approach often outperforms the "plug-in" approach of estimating a predictor from the reference class and then playing it [Merhav and Feder, 1998]. A similar Bayesian mixture probability assignment in the contextual probability assignment problem was used by Bhatt and Kim [2021], where the covering over the VC function class was obtained in a *data-dependent manner*. This idea of using a mixture over an empirical covering along with a so-called "add-$\beta$" probability assignment was then used by Bilodeau et al. [2021]. Combining this with the key idea of discretizing the class of functions as per the Hellinger divergence induced metric, they obtained matching rates for several interesting classes in the *realizable* case (i.e. $y_t|x_t$ generated according to a fixed unknown distribution in the reference class); see also Yang and Barron [1999] for more intuition behind usage of Hellinger coverings for stochastic data. Recent work of Wu et al. [2022a,b] has also employed an empirical covering with an add-$\beta$ probability assignment for both stochastic and adversarial adversaries.

A complementary approach, more common in the online learning literature is to study fundamental limits of sequential decision making problems non-constructively (i.e. providing bounds on the minmax regret without providing a probability assignment that achieves said regret). This sequential complexities based approach of Rakhlin et al. [2015b,a] has been employed for the log-loss by Rakhlin and Sridharan [2015] and Bilodeau et al. [2020]; however the latter suggests that sequential complexities might not fully capture the log-loss problem.

Smoothed analysis, initiated by Spielman and Teng [2004] for the study of efficiency of algorithms such as the simplex method, has recently shown to be effective in circumventing both statistical and computatonal lower bounds in online learning for classification and regression Haghtalab et al. [2021], Rakhlin et al. [2011], Block et al. [2022], Haghtalab et al. [2020], Block and Simchowitz [2022]. This line of work establishes that smoothed analysis is a viable line of attack to construct statistically and computationally efficient algorithms for sequential decision making problems and has developed tools such as the coupling lemma to understand smoothed sequential decision-making. But handling the unbounded and non-Lipschitz log-loss and MLE oracle is the main conceptual novelty introduced in this work.

Due to the fundamental nature of the problem, the notion of computational efficiency for sequential probability assignment and the closely related problem of portfolio selection has been considered in the literature. Kalai and Vempala [2002] presents an efficient implementation of Cover's universal portfolio algorithm using techniques from Markov chain Monte Carlo. Recently, there has been a flurry of interest in using follow the regularized leader (FTRL) type techniques to achieve low regret and low complexity simultaneously [Luo et al., 2018, Zimmert et al., 2022, Jézéquel et al., 2022], see also Van Erven et al. [2020] and the references within. However, none of these methods consider the contextual version of the problem and are considerably different from the oracle-efficient approach. On the other hand, work studying portfolio selection with contexts [Cover and Ordentlich, 1996, Cross and Barron, 2003, Györfi et al., 2006, Bhatt et al., 2023] does not take oracle-efficiency into account.

**Concurrent Work:** Wu et al. [2023] also study the problem of sequential probability assignment (and general mixable losses) and for VC classes achieve the optimal regret of $O(d \log(T/\sigma))$. In addition to the smooth adversaries, they also studied general models capturing the setting where the base measures are not known. They work primarily in the information theoretical setting and do not present any results regarding efficient algorithms. A major contribution of our work is the study of oracle-efficient online learning for the log-loss, which has not been considered by Wu et al. [2023] Indeed, to the best of knowledge, oracle efficiency has not been considered at all for losses such as the log loss even for worst-case settings. Considering oracle-efficiency for log-loss is an important and natural line of research as the ERM oracle corresponds to the maximum likelihood estimator (MLE) that is commonly used in practice. Our paper presents sublinear regret oracle efficient algorithms for the log loss (which does not have an analog in Wu et al. [2023]).

# B Deferred Proof from Section 3

In order to obtain an upper bound on $\mathcal{R}_T(\mathcal{F}, \sigma)$ in terms of $\underline{\mathcal{R}}_T^{kT}(\mathcal{F})$ for some $k$, we will consider (2) and proceed inductively. The main idea is to note that since $\mathcal{D}_i$ is $\sigma$-smoothed, conditioned on the history thus far, we can invoke the coupling lemma given in Theorem 2.1.

For the sake of illustration, first consider the simple case of $T = 1$. Let $X_1, Z_1 \ldots Z_k$ denote the coupling alluded to in Theorem 2.1. Recall that $X_1 \sim \mathcal{D}_1$ and $Z_{1:k} \sim \mu^k$. Defining the event $E_1 := \{X_1 \in Z_{1:k}\}$, we have

$$\mathcal{R}_1(\mathcal{F}, \mathscr{D}) = \mathbb{E}_{X_1 \sim \mathcal{D}_1} \inf_{a_1} \sup_{y_1} \mathcal{R}_1(\mathcal{F}, X_1, y_1, a_1)$$

$$= \mathbb{E}_{X_1, Z_{1:k}} \left[ \inf_{a_1} \sup_{y_1} \mathcal{R}_1(\mathcal{F}, X_1, y_1, a_1) \right]$$

$$= \mathbb{E}_{X_1, Z_{1:k}} \left[ \mathbb{1}\{E_1\} \inf_{a_1} \sup_{y_1} \mathcal{R}_1(\mathcal{F}, X_1, y_1, a_1) \right]$$

$$+ \mathbb{E}_{X_1, Z_{1:k}} \left[ \mathbb{1}\{E_1^C\} \inf_{a_1} \sup_{y_1} \mathcal{R}_1(\mathcal{F}, X_1, y_1, a_1) \right]$$

$$\leq \mathbb{E}_{X_1, Z_{1:k}} \left[ \mathbb{1}\{E_1\} \inf_{a_1} \sup_{y_1} \mathcal{R}_1(\mathcal{F}, X_1, y_1, a_1) \right] + \mathbb{P}(E_1^c) \tag{3}$$

$$\leq \mathbb{E}_{Z_{1:k}} \left[ \max_{X_1 \in Z_{1:k}} \inf_{a_1} \sup_{y_1} \mathcal{R}_1(\mathcal{F}, X_1, y_1, a_1) \right] + (1-\sigma)^k \tag{4}$$

$$= \underline{\mathcal{R}}_T^{kT}(\mathcal{F}) + (1-\sigma)^k, \tag{5}$$

where (3) uses that $\inf_{a_1} \sup_{y_1} \mathcal{R}_1(\mathcal{F}, X_1, y_1, a_1) \leq 1$ [4], (4) follows by the coupling lemma and (5) follows from the definition of transductive learning regret. The next step is to generalize this to arbitrary $T$. The key aspect that makes this possible is that for all $t \leq T$, we have $D_t \in \Delta_\sigma(\mu)$, even conditioned on the past, allowing us to apply the coupling lemma. Furthermore, we need that $\mathcal{R}_T \leq T$ for arbitrary sequences which is indeed guaranteed for reasonable losses such as the log-loss as noted above.

We now move to general case. We will prove this inductively. Assume that we have used the coupling lemma till time $t-1$ and replaced the samples from the smooth distributions with samples from the uniform. That is assume the induction hypothesis, for time $t$ as

$$\mathcal{R}_T \leq \mathbb{E}_{\{Z_{1:k}\} \sim \mu} \max_{X_1 \in Z_1^k} \inf_{a_1} \sup_{y_1} \ldots \sup_{\mathcal{D}_t} \mathbb{E}_{X_t \sim \mathcal{D}_t} \inf_{a_t} \sup_{y_t} \ldots$$

$$\ldots \sup_{\mathcal{D}_T} \mathbb{E}_{X_T \sim \mathcal{D}_T} \inf_{a_T} \sup_{y_T} \mathcal{R}(\mathcal{F}, X_{1:T}, y_{1:T}, a_{1:T}) + T(t-1)(1-\sigma)^k$$

Using the coupling lemma, we have that there exists a coupling $\Pi_t$ such that $X_t, Z_{t,1} \ldots Z_{t,k} \sim \Pi_t$ and an event $E_t = \left\{ X_t \in \{Z_{t,1} \ldots Z_{t,k}\} \right\}$ that occurs with probability $1 - (1-\sigma)^k$. Using $Z_t := \{Z_{t,1} \ldots Z_{t,k}\}$ we have

$$\mathbb{E}_{Z_1 \sim \mu^k} \max_{X_1 \in Z_1} \inf_{a_1} \sup_{y_1} \ldots \sup_{\mathcal{D}_t} \mathbb{E}_{X_t \sim \mathcal{D}_t} \inf_{a_t} \sup_{y_t} \ldots \sup_{\mathcal{D}_T} \mathbb{E}_{X_T \sim \mathcal{D}_T} \inf_{a_T} \sup_{y_T} \mathcal{R}(\mathcal{F}, X_{1:T}, y_{1:T}, a_{1:T})$$

$$\leq \mathbb{E}_{Z_1 \sim \mu^k} \max_{X_1 \in Z_1} \inf_{a_1} \sup_{y_1} \ldots \sup_{\mathcal{D}_t} \mathbb{E}_{X_t, Z_t \sim \Pi_t} \inf_{a_t} \sup_{y_t} \ldots \sup_{\mathcal{D}_T} \mathbb{E}_{X_T \sim \mathcal{D}_T} \inf_{a_T} \sup_{y_T} \mathcal{R}(\mathcal{F}, X_{1:T}, y_{1:T}, a_{1:T})$$

$$\leq \mathbb{E}_{Z_1 \sim \mu^k} \max_{X_1 \in Z_1} \inf_{a_1} \sup_{y_1} \ldots \sup_{\mathcal{D}_t} \mathbb{E}_{X_t, Z_t \sim \Pi_t} \left[ \mathbb{1}[E_t] \left( \inf_{a_t} \sup_{y_t} \ldots \sup_{\mathcal{D}_T} \mathbb{E}_{X_T \sim \mathcal{D}_T} \inf_{a_T} \sup_{y_T} \mathcal{R}(\mathcal{F}, X_{1:T}, y_{1:T}, a_{1:T}) \right) \right]$$

$$+ \mathbb{E}_{Z_1 \sim \mu^k} \max_{X_1 \in Z_1} \inf_{a_1} \sup_{y_1} \ldots \sup_{\mathcal{D}_t} \mathbb{E}_{X_t, Z_t \sim \Pi_t} \left[ \mathbb{1}[E_t^c] \left( \inf_{a_t} \sup_{y_t} \ldots \sup_{\mathcal{D}_T} \mathbb{E}_{X_T \sim \mathcal{D}_T} \inf_{a_T} \sup_{y_T} \mathcal{R}(\mathcal{F}, X_{1:T}, y_{1:T}, a_{1:T}) \right) \right]$$

---

[4]This holds for the log-loss by using the trivial strategy of using a uniform probability assignment at each step.

$$\leq \mathop{\mathbb{E}}_{Z_1 \sim \mu^k} \max_{X_1 \in Z_1} \inf_{a_1} \sup_{y_1} \dots \sup_{\mathcal{D}_t} \mathop{\mathbb{E}}_{X_t, Z_t \sim \Pi_t} \left[ \mathbb{1}[E_t] \left( \inf_{a_t} \sup_{y_t} \dots \sup_{\mathcal{D}_T} \mathop{\mathbb{E}}_{X_T \sim \mathcal{D}_T} \inf_{a_T} \sup_{y_T} \mathcal{R}(\mathcal{F}, X_{1:T}, y_{1:T}, a_{1:T}) \right) \right]$$

$$+ T (1 - \sigma)^k$$

$$\leq \mathop{\mathbb{E}}_{Z_1 \sim \mu^k} \max_{X_1 \in Z_1} \inf_{a_1} \sup_{y_1} \dots \sup_{\mathcal{D}_t} \mathop{\mathbb{E}}_{Z_t \sim \Pi_t} \max_{X_t \in Z_t} \left( \inf_{a_t} \sup_{y_t} \dots \sup_{\mathcal{D}_T} \mathop{\mathbb{E}}_{X_T \sim \mathcal{D}_T} \inf_{a_T} \sup_{y_T} \mathcal{R}(\mathcal{F}, X_{1:T}, y_{1:T}, a_{1:T}) \right)$$

$$+ T (1 - \sigma)^k$$

$$= \mathop{\mathbb{E}}_{Z_1 \sim \mu^k} \max_{X_1 \in Z_1} \inf_{a_1} \sup_{y_1} \dots \mathop{\mathbb{E}}_{Z_t \sim \mu^k} \max_{X_t \in Z_t} \left( \inf_{a_t} \sup_{y_t} \dots \sup_{\mathcal{D}_T} \mathop{\mathbb{E}}_{X_T \sim \mathcal{D}_T} \inf_{a_T} \sup_{y_T} \mathcal{R}(\mathcal{F}, X_{1:T}, y_{1:T}, a_{1:T}) \right)$$

$$+ T (1 - \sigma)^k$$

Combining with the induction hypothesis, gives us the induction hypothesis for the next $t$ as required. The desired result follows by upper bounding the average with the supremum over all subsets of size $kT$.

## C  Proof of Theorem 3.2

First, recall the notion of the global sequential covering for a class Wu et al. [2022b].

**Definition C.1** (Global Sequential Covering Wu et al. [2022b]). For any class $\mathcal{F}$, we say that $\mathcal{F}'_\alpha \subset \mathcal{X}^* \to [0, 1]$ is a global sequential $\alpha$-covering of $\mathcal{F}$ at scale $T$ if for any sequence $x_{1:T}$ and $h \in \mathcal{F}$, there is a $h' \in \mathcal{F}'$ such that for all $i$,

$$\left| h(x_i) - h'(x_{1:i}) \right| \leq \alpha.$$

**Theorem C.1** (Wu et al. [2022b]). *If $\mathcal{F}'_\alpha$ is a global sequential $\alpha$-covering of $\mathcal{F}$ at scale $T$, then*

$$\mathcal{R}_T (\mathcal{F}) \leq \inf_{\alpha > 0} \left\{ 2\alpha T + \log \left| \mathcal{F}'_\alpha \right| \right\}.$$

To finish the proof note that a $\epsilon$-cover in the sense of Definition 3.1 gives a global sequential cover in the sense of Definition C.1.

## D  VC Classes

In this section, we construct a probability assignment for the case when $\mathcal{F} \subset \{\mathcal{X} \to [0, 1]\}$ is a *VC class*. To motivate this probability assignment, consider the no-context case, which is a classic problem in information theory, where the (asymptotically) optimal probability assignment is known to be the Krichevsky and Trofimov [1981] (KT) probability assignment which is a Bayesian mixture of the form

$$q_{\text{KT}}(y_{1:T}) = \int_0^1 p_\theta(y_{1:T}) w(\theta) d\theta$$

for a particular prior $w(\theta)$. This can be written sequentially as $q_{\text{KT}}(1|y_{1:t-1}) = \frac{\sum_{i=1}^{t-1} y_i + 1/2}{t - 1 + 1}$ leading to it sometimes being called the add-1/2 probability assignment; by choosing $w(\theta)$ to be $\text{Beta}(\beta, \beta)$ prior one can achieve a corresponding add-$\beta$ probability assignment. We extend the mixture idea to the contextual case. In particular, for functions $f_1, \dots, f_m \in \mathcal{F}$, one can choose a mixture probability assignment as [5]

$$\prod_{i=1}^t q(y_i | x_{1:i}, y_{1:i-1}) =: q(y_{1:t} \| x_{1:t}) = \frac{1}{m} \sum_{j=1}^m \prod_{i=1}^t \left( \frac{p_{f_j}(y_i | x_i) + \beta}{1 + 2\beta} \right).$$

---

[5] Note that once a mixture $q(y_{1:t} \| x_{1:t})$ has been defined for arbitrary $x_{1:t}, y_{1:t}$, the probability assignment at time $t$ (or equivalently, the predicted probability with which the upcoming bit is 1) can be defined as $q(1 | x_{1:t}, y_{1:t-1}) = \frac{q(y_{1:t-1} 1 \| x_{1:t})}{q(y_{1:t-1} \| x_{1:t-1})}$; in particular, this prediction depends only on the observed history $x_{1:t}, y_{1:t-1}$ and not the future $y_t$.

This is the approach employed presently with a carefully chosen $f_1, \ldots, f_m$. We remark that for VC classes this mixture approach may be extended to any *mixable* [Cesa-Bianchi and Lugosi, 2006, Chapter 3] loss.

First consider VC classes more carefully: i.e. each $f \in \mathcal{F}^{VC}$ is characterized by three things: a set $A \subseteq \mathcal{X}$, where $A \in \mathcal{A} \subset 2^{\mathcal{X}}$ with the VC dimension of the collection $\mathcal{A}$ being $d < \infty$; as well as two numbers $\theta_0, \theta_1 \in [0, 1]$. Then, we have

$$f_{A,\theta_0,\theta_1}(x) = \theta_0 \mathbb{1}\{x \in A\} + \theta_1 \mathbb{1}\{x \in A^C\}.$$

The following equivalent representation of this hypothesis class is more convenient to use. We consider each $f$ to be characterized by a tuple $f = (g, \theta_0, \theta_1)$ where

1. $\theta_0, \theta_1 \in [0, 1]$
2. $g \in \mathcal{G} \subset \{\mathcal{X} \to \{0, 1\}\}$.

In other words, $g$ belongs to a class $\mathcal{G}$ of binary functions—this is simply the class of functions $\{x \mapsto \mathbb{1}\{x \notin A\} \,|\, A \in \mathcal{A}\}$ in the original notation; so that clearly $\text{VCdim}(\mathcal{G}) = d$. Then, we have $p_f(\cdot|x) = p_{g,\theta_0,\theta_1}(\cdot|x) = \text{Bernoulli}(\theta_0)$ if $g(x) = 0$; and $p_{g,\theta_0,\theta_1}(\cdot|x) = \text{Bernoulli}(\theta_1)$ otherwise.

Recalling the definition of regret against a particular $f = (g, \theta_0, \theta_1)$ for a sequential probability assignment strategy $\mathcal{Q} = \{q(\cdot|x_{1:t}, y_{1:t-1})\}_{t=1}^T$

$$\mathcal{R}_T(f, x_{1:T}, y_{1:T}, \mathcal{Q}) = \sum_{t=1}^T \log \frac{1}{q(y_t|x_{1:t}, y_{1:t-1})} - \sum_{t=1}^T \log \frac{1}{p_f(y_t|x_t)} \tag{6}$$

$$= \log \frac{p_f(y_{1:T}|x_{1:T})}{q(y_{1:T}\|x_{1:T})}$$

where $q(y_{1:T}\|x_{1:T}) := \prod_{t=1}^T q(y_t|x_{1:t}, y_{1:t-1})$.

In the smoothed analysis case, we have $X_t \sim \mathcal{D}_t$ where $\mathcal{D}_t$ for all $t$ is $\sigma$-smoothed. Recall that in this case, we are concerned with the regret

$$\mathcal{R}_T(\mathcal{F}, \sigma, \mathcal{Q}) = \max_{\mathscr{D}:\sigma\text{-smoothed}} \mathbb{E}_{X_{1:T}} \left[ \max_{y_{1:T}} \sup_{f \in \mathcal{F}} \frac{p_f(y_{1:T}|X_{1:T})}{q(y_{1:T}\|X_{1:T})} \right]$$

$$= \max_{\mathscr{D}:\sigma\text{-smoothed}} \mathbb{E}_{X_{1:T}} \left[ \max_{y_{1:T}} \sup_{g \in \mathcal{G}} \max_{\theta_0,\theta_1} \frac{p_{g,\theta_0,\theta_1}(y_{1:T}|X_{1:T})}{q(y_{1:T}\|X_{1:T})} \right].$$

### D.1 Proposed probability assignment

Let $\mu$ be the dominating measure for the $\sigma$-smoothed distribution of $X_{1:T}$. Let $g_1, \ldots, g_{m_\epsilon} \in \mathcal{G}$ be an $\epsilon$-cover of the function class $\mathcal{G}$ under the metric $\delta_\mu(g_1, g_2) = \Pr_{X \sim \mu}(g_1(X) \neq g_2(X))$. The following lemma bounds $m_\epsilon$.

**Lemma D.1** (Covering number of VC classes under the metric $\delta$, Vershynin [2018])**.**

$$m_\epsilon \leq \left(\frac{1}{\epsilon}\right)^{cd}$$

*for an absolute constant $c$.*

Following the idea of using a mixture probability assignment, we take a uniform mixture over $g_1, \ldots, g_{m_\epsilon}$ and $\theta_0, \theta_1$ so that

$$q(y_{1:t}\|x_{1:t}) = \frac{1}{m_\epsilon} \sum_{i=1}^{m_\epsilon} \int_0^1 \int_0^1 p_{g_i,\theta_0,\theta_1}(y_{1:t}|x_{1:t}) d\theta_0 d\theta_1$$

and consequently the sequential probability assignment (or equivalently, the probability assigned to 1) is

$$q(1|x_{1:t}, y_{1:t-1}) = \frac{q(y_{1:t-1}1\|x_{1:t})}{q(y_{1:t-1}\|x_{1:t-1})}.$$

One can observe that $q(0|x_{1:t}, y_{1:t-1}), q(1|x_{1:t}, y_{1:t-1}) > 0$ and $q(0|x_{1:t}, y_{1:t-1}) + q(1|x_{1:t}, y_{1:t-1}) = 1$ so that $q$ is a legitimate probability assignment. Let the strategy induced by this uniform mixture be called $\mathcal{Q}^{VC}$.

## D.2 Analysis of $\mathcal{Q}^{\mathrm{VC}}$ for smoothed adversaries

We note from (6) that for the $\mathcal{Q}^{\mathrm{VC}}$ as defined in the last section, we have

$$\mathcal{R}_T((g^*, \theta_0^*, \theta_1^*), x_{1:T}, y_{1:T}, \mathcal{Q}^{\mathrm{VC}}) = \log m_\epsilon + \log \frac{p_{g^*, \theta_0^*, \theta_1^*}(y_{1:T}|x_{1:T})}{\sum_{i=1}^{m_\epsilon} \int_0^1 \int_0^1 p_{g_i, \theta_0, \theta_1}(y_{1:T}|x_{1:T})d\theta_0 d\theta_1}$$

$$\leq \log m_\epsilon + \log \frac{p_{g^*, \theta_0^*, \theta_1^*}(y_{1:T}|x_{1:T})}{\int_0^1 \int_0^1 p_{g_{i^*}, \theta_0, \theta_1}(y_{1:T}|x_{1:T})d\theta_0 d\theta_1} \quad (7)$$

where $g_{i^*} \in \{g_1, \ldots, g_{m_\epsilon}\}$ is the function $i^* \in [m]$ that minimizes the Hamming distance between the binary strings $(g_{i^*}(x_1), \ldots, g_{i^*}(x_T))$ and $(g^*(x_1), \ldots, g^*(x_T))$.

We now take a closer look at the second term of (7). Firstly, note that for any $(g, \theta_0, \theta_1)$ we have $p_{g,\theta_0,\theta_1}(y_{1:T}|x_{1:T}) =$

$$\prod_{t=1}^T p_{g,\theta_0,\theta_1}(y_t|x_t) = \prod_{t=1}^T \theta_{g(x_t)}^{y_t}(1-\theta_{g(x_t)})^{1-y_t} = \prod_{t:g(x_t)=0} \theta_0^{y_t}(1-\theta_0)^{1-y_t} \prod_{t:g(x_t)=1} \theta_1^{y_t}(1-\theta_1)^{1-y_t}$$

$$= \theta_0^{k_0(g;x_{1:T},y_{1:T})}(1-\theta_0)^{n_0(g;x_{1:T})-k_0(g;x_{1:T},y_{1:T})}$$

$$\theta_1^{k_1(g;x_{1:T},y_{1:T})}(1-\theta_1)^{n_1(g;x_{1:T})-k_1(g;x_{1:T},y_{1:T})},$$

where for $j \in \{0,1\}$

$$k_j(g; x_{1:T}, y_{1:T}) = |\{t : y_t = 1, g(x_t) = j\}|$$
$$n_0(g; x_{1:T}) = |\{t : g(x_t) = j\}|.$$

Next, we note that for any $g \in \mathcal{G}$

$$\int_0^1 \int_0^1 p_{g,\theta_0,\theta_1}(y_{1:T}|x_{1:T})d\theta_0 d\theta_1$$

$$= \left( \int_0^1 \theta_0^{k_0(g;x_{1:T},y_{1:T})}(1-\theta_0)^{n_0(g;x_{1:T})-k_0(g;x_{1:T},y_{1:T})}d\theta_0 \right) \cdot$$

$$\left( \int_0^1 \theta_1^{k_1(g;x_{1:T},y_{1:T})}(1-\theta_1)^{n_1(g;x_{1:T})-k_0(g;x_{1:T},y_{1:T})}d\theta_1 \right)$$

$$= \frac{1}{\binom{n_0(g;x_{1:T})}{k_0(g;x_{1:T},y_{1:T})}(n_0(g;x_{1:T})+1)} \frac{1}{\binom{n_1(g;x_{1:T})}{k_1(g;x_{1:T},y_{1:T})}(n_1(g;x_{1:T})+1)} \quad (8)$$

$$\geq \frac{1}{n^2 \binom{n_0(g;x_{1:T})}{k_0(g;x_{1:T},y_{1:T})}\binom{n_1(g;x_{1:T})}{k_1(g;x_{1:T},y_{1:T})}}$$

where (8) follows from properties of the Laplace probability assignment (or that of the Beta/Gamma functions), captured by Lemma D.2.

**Lemma D.2.** *For $k \leq n \in \mathbb{N}$,*

$$\int_0^1 t^k(1-t)^{n-k}dt = \frac{\Gamma(k+1)\Gamma(n-k+1)}{\Gamma(n+2)} = \frac{1}{(n+1)\binom{n}{k}}$$

*where $\Gamma(\cdot)$ represents the Gamma function.*

Putting this back into (7) (and rearranging), we have

$$\mathcal{R}_T((g^*, \theta_0^*, \theta_1^*), x_{1:T}, y_{1:T}, \mathcal{Q}^{\mathrm{VC}}) - \log m_\epsilon - 2\log n$$

$$\leq \sum_{j \in \{0,1\}} \log \left( \binom{n_j(g_{i^*};x_{1:T})}{k_j(g_{i^*};x_{1:T},y_{1:T})}(\theta_j^*)^{k_j(g^*;x_{1:T},y_{1:T})}(1-\theta_j^*)^{n_j(g^*;x_{1:T})-k_j(g^*;x_{1:T},y_{1:T})} \right)$$

$$= \sum_{j \in \{0,1\}} \log \left( \binom{n_j(g_{i^*};x_{1:T})}{k_j(g_{i^*};x_{1:T},y_{1:T})} \binom{n_j(g^*;x_{1:T})}{k_j(g^*;x_{1:T},y_{1:T})}^{-1} \right).$$

$$\binom{n_j(g^*; x_{1:T})}{k_j(g^*; x_{1:T}, y_{1:T})} (\theta_j^*)^{k_j(g^*; x_{1:T}, y_{1:T})} (1 - \theta_j^*)^{n_j(g^*; x_{1:T}, y_{1:T}) - k_j(g^*; x_{1:T}, y_{1:T})} \Bigg)$$

$$\leq \sum_{j \in \{0,1\}} \log \left( \binom{n_j(g_{i^*}; x_{1:T})}{k_j(g_{i^*}; x_{1:T}, y_{1:T})} \binom{n_j(g^*; x_{1:T})}{k_j(g^*; x_{1:T}, y_{1:T})}^{-1} \right) \tag{9}$$

where (9) follows since for any natural numbers $k \leq n$ and $\theta \in [0,1]$ we have $\binom{n}{k} \theta^k (1-\theta)^{n-k} \leq 1$. Now, note that

$$\log \frac{\binom{n}{k}}{\binom{n'}{k'}} = \log \frac{n!}{n'!} + \log \frac{k'!}{k!} + \log \frac{(n'-k')!}{(n-k)!}$$

$$\leq \log \frac{(n' + |n - n'|)!}{n'!} + \log \frac{(k + |k - k'|)!}{k!} + \log \frac{((n-k) + |n - n'| + |k - k'|)!}{(n-k)!}$$

If $|k - k'|, |n - n'| \leq \delta$, and $\max\{n, n'\} \leq N$ then by for example [Bhatt and Kim, 2021, Proposition 6] we have that

$$\log \frac{\binom{n}{k}}{\binom{n'}{k'}} \leq 2\delta \log(n' + 2\delta) + 2\delta \log(k + 2\delta) + 4\delta \log((n-k) + 4\delta)$$

$$\leq 16\delta \log N. \tag{10}$$

We now wish to use this bound in (9). For this, we will recall the definitions of $n_0(g; x_{1:T})$ and $k_0(g; x_{1:T}, y_{1:T})$ for a particular function $g$ and observe that for two functions $g, g'$ we have that both $|n_0(g; x_{1:T}) - n_0(g'; x_{1:T})|, |k_0(g; x_{1:T}, y_{1:T}) - k_0(g'; x_{1:T}, y_{1:T})| \leq d_H(g(x_{1:T}), g'(x_{1:T}))$ where $d_H(\cdot, \cdot)$ denotes the Hamming distance and $g(x_{1:T}) := (g(x_1), \ldots, g(x_T)) \in \{0, 1\}^T$. Thus, by using (10) in (9) with $\delta = d_H(g^*(x_{1:T}), g_{i^*}(x_{1:T}))$, $N = T$, we get

$$\mathcal{R}_T((g^*, \theta_0^*, \theta_1^*), x_{1:T}, y_{1:T}, \mathcal{Q}^{\text{VC}}) \leq \log m_\epsilon + 2\log T + 32 d_H(g^*(x_{1:T}), g_{i^*}(x_{1:T})) \log T. \tag{11}$$

Note that (11) has effectively removed any dependence on $y, \theta_0^*, \theta_1^*$. We then have for some absolute constant $C$, (recalling the definition of $i^*$ and $\mathcal{F}$ from earlier)

$$\mathcal{R}_T(\mathcal{F}, \sigma, \mathcal{Q}^{\text{VC}}) \leq C \log m_\epsilon + C \log T \max_{\mathcal{D}: \sigma\text{-smoothed}} \mathbb{E} \left[ \sup_{g^* \in \mathcal{G}} \min_{i \in [m_\epsilon]} d_H(g^*(X_{1:T}), g_i(X_{1:T})) \right]. \tag{12}$$

Finally, we can control the last term in (12) by the following result, which follows from the coupling lemma and variance sensitive upper bounds on suprema over VC classes.

**Lemma D.3** (Lemma 3.3 of Haghtalab et al. [2021]).

$$\mathbb{E} \left[ \sup_{g^* \in \mathcal{G}} \min_{i \in [m]} d_H(g^*(X_{1:T}), g_i(X_{1:T})) \right] \leq \sqrt{\frac{\epsilon}{\sigma} T \log T d \log\left(\frac{1}{\epsilon}\right)} + T \log T \frac{\epsilon}{\sigma}$$

Plugging the above into (12) and taking $\epsilon = \frac{\sigma}{T^2}$ gives us

$$\mathcal{R}_T(\mathcal{F}, \sigma, \mathcal{Q}^{\text{VC}}) \leq O\left(d \log\left(\frac{T}{\sigma}\right)\right).$$

# E  Proof of Lemma 4.2

*Proof.* Note that this proof holds for general loss functions. Let $\mathcal{R}_T$ denote the regret.

$$\mathcal{R}_T \leq \mathbb{E} \left[ \sum_{i=1}^{T} \ell(h_t, s_t) - \inf_{h \in \mathcal{F}} \sum_{i=1}^{T} \ell(h, s_t) \right]$$

$$= \mathbb{E} \left[ \sum_{i=1}^{T} \ell(h_t, s_t) - \sum_{t=1}^{T} \ell(h_{t+1}, s_t) + \sum_{t=1}^{T} \ell(h_{t+1}, s_t) - \inf_{h \in \mathcal{F}} \sum_{i=1}^{T} \ell(h, s_t) \right]$$

$$= \mathbb{E}\left[\sum_{i=1}^{T}\ell(h_t, s_t) - \sum_{t=1}^{T}\ell(h_{t+1}, s_t)\right] + \mathbb{E}\left[\sum_{t=1}^{T}\ell(h_{t+1}, s_t) - \inf_{h\in\mathcal{F}}\sum_{i=1}^{T}\ell(h, s_t)\right]$$

Let us focus on the second term.

$$\mathbb{E}\left[\sum_{t=1}^{T}\ell(h_{t+1}, s_t) - \inf_{h\in\mathcal{F}}\sum_{i=1}^{T}\ell(h, s_t)\right]$$

$$\leq \mathbb{E}\left[\sum_{t=1}^{T}\ell(h_{t+1}, s_t) - \inf_{h\in\mathcal{F}_\alpha}\sum_{i=1}^{T}\ell(h, s_t) + \inf_{h\in\mathcal{F}_\alpha}\sum_{i=1}^{T}\ell(h, s_t) - \inf_{h\in\mathcal{F}}\sum_{i=1}^{T}\ell(h, s_t)\right]$$

$$\leq 2\alpha T + \mathbb{E}\left[\sum_{t=1}^{T}\ell(h_{t+1}, s_t) - \inf_{h\in\mathcal{F}_\alpha}\sum_{i=1}^{T}\ell(h, s_t)\right] \tag{13}$$

$$\leq 2\alpha T + \mathbb{E}\left[\sum_{t=1}^{N}\ell(h_t, \tilde{s}_t) - \ell(h^*, \tilde{s}_t)\right] \tag{14}$$

$$\leq 2\alpha T + \mathbb{E}\left[\sup_{h\in\mathcal{F}_\alpha}\sum_{t=1}^{N}\ell(h, \tilde{s}_t) - \ell(h^*, \tilde{s}_t)\right]$$

where $h^* = \inf_{h\in\mathcal{F}_\alpha}\sum_{i=1}^{T}\ell(h, s_t)$. (13) follows by comparing the optimal of the truncated class with the whole class, see [Cesa-Bianchi and Lugosi, 2006, Lemma 9.5]. (14) follows from the Be-the-leader lemma Cesa-Bianchi and Lugosi [2006]. □

## F    Proof of Lemma 4.3

Denote by $R^{(t)} = (N^{(t)}, \{\tilde{s}_i\}_{i\in N^{(t)}})$ the fresh randomness generated at the beginning of time $t$, which is independent of $\{s_\tau\}_{\tau<t}$ generated by the adversary. Let $\mathcal{Q}_t$ be the distribution of the learner's action $h_t \in \mathcal{H}$ in Algorithm 1, Formally,

$$r^t(x) = \sum_{i=1}^{N^{(t+1)}} \tilde{y}_i^{(t+1)} \cdot \mathbf{1}(\tilde{x}_i^{(t+1)} = x) + \sum_{\tau=1}^{t} y_\tau \cdot \mathbf{1}(x_\tau = x).$$

Let $\mathcal{P}^t$ be the distribution of $r^t$. The reason why we introduce this notion is that $h_t$ in Algorithm 1 only depends on the vector $r^{t-1}$.

The main step in the proof is to introduce an independent sample from the distribution $\mathcal{D}_t$ in order to decouple the dependence of the distribution $\mathcal{Q}_{t+1}$ on the test point $s_t$.

$$\mathbb{E}_{s_t\sim\mathcal{D}_t}\mathbb{E}_{h_t\sim\mathcal{Q}_t}[\ell(h_t, s_t)] - \mathbb{E}_{s_t\sim\mathcal{D}_t}\mathbb{E}_{h_{t+1}\sim\mathcal{Q}_{t+1}}[\ell(h_{t+1}, s_t)]$$

$$= \mathbb{E}_{s_t\sim\mathcal{D}_t}\mathbb{E}_{h_t\sim\mathcal{Q}_t}[\ell(h_t, s_t)] - \mathbb{E}_{s_t,s_t'\sim\mathcal{D}_t}\mathbb{E}_{h_{t+1}\sim\mathcal{Q}_{t+1}}[\ell(h_{t+1}, s_t')]$$

$$+ \mathbb{E}_{s_t,s_t'\sim\mathcal{D}_t}\mathbb{E}_{h_{t+1}\sim\mathcal{Q}_{t+1}}[\ell(h_{t+1}, s_t')] - \mathbb{E}_{s_t\sim\mathcal{D}_t}\mathbb{E}_{h_{t+1}\sim\mathcal{Q}_{t+1}}[\ell(h_{t+1}, s_t)] \tag{15}$$

$$= \mathbb{E}_{s_t'\sim\mathcal{D}_t}\mathbb{E}_{h_t\sim\mathcal{Q}_t}[\ell(h_t, s_t')] - \mathbb{E}_{s_t'\sim\mathcal{D}_t}\mathbb{E}_{h_{t+1}\sim\mathbb{E}_{s_t\sim\mathcal{D}_t}[\mathcal{Q}_{t+1}]}[\ell(h_{t+1}, s_t')] \tag{16}$$

$$+ \mathbb{E}_{s_t,s_t'\sim\mathcal{D}_t}\mathbb{E}_{h_{t+1}\sim\mathcal{Q}_{t+1}}[\ell(h_{t+1}, s_t')] - \mathbb{E}_{s_t\sim\mathcal{D}_t}\mathbb{E}_{h_{t+1}\sim\mathcal{Q}_{t+1}}[\ell(h_{t+1}, s_t)]$$

where we get (15) by adding and subtracting the middle term corresponding to evaluating the loss on an independent sample $s_t'$ and (16) by observing that $s_t$ and $s_t'$ are equally distributed. Since the second term is the same in the required equation, we can focus on the first term.

$$\mathbb{E}_{h_t\sim\mathcal{Q}_t}\left[\mathbb{E}_{s_t'\sim\mathcal{D}_t}[\ell(h_t, s_t')]\right] - \mathbb{E}_{h_{t+1}\sim\widetilde{\mathcal{Q}_{t+1}}}\left[\mathbb{E}_{s_t'\sim\mathcal{D}_t}[\ell(h_{t+1}, s_t')]\right]. \tag{17}$$

Here we use the notation $\widetilde{\mathcal{Q}_{t+1}} = \mathbb{E}_{s_t \sim \mathcal{D}_t}[\mathcal{Q}_{t+1}]$ for the mixture distribution. In order to bound this, we look a variational interpretation of the $\chi^2$ distance between two distributions $P$ and $Q$.

**Lemma F.1** (Hammersley–Chapman–Robbins bound). *For any pair of measures $P$ and $Q$ and any measurable function $h : \mathcal{X} \to \mathbb{R}$, we have*

$$\left| \mathbb{E}_{X \sim P}[h(X)] - \mathbb{E}_{X \sim Q}[h(X)] \right| \leq \sqrt{\chi^2(P, Q) \cdot \mathrm{Var}_{X \sim Q}(h(X))}$$
$$\leq \sqrt{\frac{1}{2} \chi^2(P, Q) \cdot \mathbb{E}_{X, X' \sim Q}(h(X) - h(X'))^2}.$$

Applying this to (17), we get

$$\mathbb{E}_{h_t \sim \mathcal{Q}_t}\left[ \mathbb{E}_{s'_t \sim \mathcal{D}_t}[\ell(h_t, s'_t)] \right] - \mathbb{E}_{h_{t+1} \sim \widetilde{\mathcal{Q}_{t+1}}}\left[ \mathbb{E}_{s'_t \sim \mathcal{D}_t}[\ell(h_{t+1}, s'_t)] \right]$$
$$\leq \sqrt{\frac{1}{2} \chi^2(\mathbb{E}_{s_t \sim \mathcal{D}_t}[\mathcal{Q}_{t+1}], \mathcal{Q}_t) \cdot \mathbb{E}_{h_t, h'_t \sim \mathcal{Q}_t}\left( \mathbb{E}_{s_t \sim \mathcal{D}_t}[\ell(h_t, s_t) - \ell(h'_t, s_t)] \right)^2}.$$

as required. As noted before, for the particular use in our analysis a simpler version of the lemma similar to Haghtalab et al. [2022] suffices but we include the general version since we believe such a version is useful in providing improved regret bounds for the problem.

# G  Upper Bounding $\chi^2$ Distance: Proof of Lemma 4.4

In this section, we will focus on bounding the $\chi^2$ distance between the distribution of actions at time steps. The reasoning in this section closely follows Haghtalab et al. [2022]. We reproduce it here for completeness.

We assume that $\mathcal{X}$ is discrete. Define

$$n_0(x) = \sum_{i=1}^{N} \mathbf{1}(\widetilde{x}_i = x, \widetilde{y}_i = 0) \quad \text{and} \quad n_1(x) = \sum_{i=1}^{N} \mathbf{1}(\widetilde{x}_i = x, \widetilde{y}_i = 1).$$

As each $\widetilde{x}_i$ is uniformly distributed on $\mathcal{X}$ and $\widetilde{y}_i \sim \mathcal{U}(\{0, 1\})$, by the subsampling property of the Poisson distribution, the $2|\mathcal{X}|$ random variables $\{n_0(x), n_1(x)\}_{x \in \mathcal{X}}$ are i.i.d. distributed as $\mathrm{Poi}(n/2|\mathcal{X}|)$.

Since the historic data is only a translation, it suffices to consider the distributions at time $t = 0$ and $t = 1$. Let $n_0^1(x) = n_0(x) + \mathbf{1}(x_1 = x, y_1 = 0)$ with $n_1^1$ definied similarly. Let $P$ and $Q$ be the probability distributions of $\{n_0(x), n_1(x)\}_{x \in \mathcal{X}}$ and $\{n_0^1(x), n_1^1(x)\}_{x \in \mathcal{X}}$, respectively. Note that the output of the oracle depends only on this vector and thus by the data processing inequality it suffices to bound $\chi^2(P, Q)$.

Note that the distribution $P$ is a product Poisson distribution:

$$P(\{n_0(x), n_1(x)\}) = \prod_{x \in \mathcal{X}} \prod_{y \in \{0, 1\}} \mathbb{P}(\mathrm{Poi}(n/2|\mathcal{X}|) = n_y(x)).$$

As for the distribution $Q$, it could be obtained from $P$ in the following way: the smooth adversary draws $x^\star \sim \mathcal{D}$, independent of $\{n_0(x), n_1(x)\}_{x \in \mathcal{X}} \sim P$, for some $\sigma$-smooth distribution $\mathcal{D} \in \Delta_\sigma(\mathcal{X})$. He then chooses a label $y^\star = y(x^\star) \in \{0, 1\}$ as a function of $x^\star$, and sets

$$n_{y(x^\star)}^1(x^\star) = n_{y(x^\star)}(x^\star) + 1, \quad \text{and} \quad n_y^1(x) = n_y(x), \quad \forall (x, y) \neq (x^\star, y(x^\star)).$$

Consequently, given a $\sigma$-smooth distribution $\mathcal{D}$ and a labeling function $y : \mathcal{X} \to \{0, 1\}$ used by the adversary, the distribution $Q$ is a mixture distribution $Q = \mathbb{E}_{x^\star \sim \mathcal{D}^\mathcal{X}}[Q_{x^\star}]$, with

$$Q_{x^\star}(\{n_0^1(x), n_1^1(x)\}) = \mathbb{P}(\mathrm{Poi}(n/2|\mathcal{X}|) = n_{y(x^\star)}(x^\star) - 1) \times \prod_{(x, y) \neq (x^\star, y(x^\star))} \mathbb{P}(\mathrm{Poi}(n/2|\mathcal{X}|) = n_y(x)).$$

We will use the Ingster method to control the $\chi^2$ between the mixture distribution $Q$ and the base distribution $P$.

**Lemma G.1** (Ingster's $\chi^2$ method). *For a mixture distribution $\mathbb{E}_{\theta \sim \pi}[Q_\theta]$ and a generic distribution $P$, the following identity holds:*

$$\chi^2\left(\mathbb{E}_{\theta \sim \pi}[Q_\theta], P\right) = \mathbb{E}_{\theta, \theta' \sim \pi}\left[\mathbb{E}_{x \sim P}\left(\frac{Q_\theta(x)Q_{\theta'}(x)}{P(x)^2}\right)\right] - 1,$$

*where $\theta'$ is an independent copy of $\theta$.*

Let $x_1^\star, x_2^\star$ be an arbitrary pair of instance. Using the closed-form expressions of distributions $P$ and $Q_{x^\star}$, it holds that

$$\frac{Q_{x_1^\star} Q_{x_2^\star}}{P^2} = \frac{2|\mathcal{X}|n_{y(x_1^\star)}(x_1^\star)}{n} \cdot \frac{2|\mathcal{X}|n_{y(x_2^\star)}(x_2^\star)}{n}.$$

Using the fact that $\{n_0(x), n_1(x)\}_{x \in \mathcal{X}}$ are i.i.d. distributed as $\mathrm{Poi}(n/2|\mathcal{X}|)$ under $P$, we have

$$\mathbb{E}_{\{n_0(x), n_1(x)\} \sim P}\left(\frac{Q_{x_1^\star}(\{n_0^1(x), n_1^1(x)\}) Q_{x_2^\star}(\{n_0^1(x), n_1^1(x)\})}{P(\{n_0(x), n_1(x)\})^2}\right) = 1 + \frac{2|\mathcal{X}|}{n} \cdot \mathbf{1}(x_1^\star = x_2^\star).$$

We will use the fact that the probability of collision between two independent draws $x_1^\star, x_2^\star \sim \mathcal{D}$ is small. That is using the Lemma G.1, we have

$$\sqrt{\frac{\chi^2(Q, P)}{2}} = \sqrt{\frac{\chi^2(\mathbb{E}_{x^\star \sim \mathcal{D}}[Q_{x^\star}], P)}{2}} = \sqrt{\frac{|\mathcal{X}|}{n} \cdot \mathbb{E}_{x_1^\star, x_2^\star \sim \mathcal{D}}[\mathbf{1}(x_1^\star = x_2^\star)]}$$

$$= \sqrt{\frac{|\mathcal{X}|}{n} \sum_{x \in \mathcal{X}} \mathcal{D}(x)^2} \leq \sqrt{\frac{|\mathcal{X}|}{n} \sum_{x \in \mathcal{X}} \mathcal{D}(x) \cdot \frac{1}{\sigma|\mathcal{X}|}} = \frac{1}{\sqrt{\sigma n}},$$

where the last inequality follows from the definition of a $\sigma$-smooth distribution.

# H   Upper Bounding Generalization Error: Proof of Lemma 4.5

The proof of the theorem is similar to the [Haghtalab et al., 2022, Section 4.2.2]. In our setting, we need to deal with general losses. We shall need the following property of smooth distributions which is a slightly strengthened version of the coupling lemma in Theorem 2.1 shown in Haghtalab et al. [2022].

**Lemma H.1.** *Let $X_1, \cdots, X_m \sim Q$ and $P$ be another distribution with a bounded likelihood ratio: $dP/dQ \leq 1/\sigma$. Then using external randomness $R$, there exists an index $I = I(X_1, \cdots, X_m, R) \in [m]$ and a success event $E = E(X_1, \cdots, X_m, R)$ such that $\Pr[E^c] \leq (1-\sigma)^m$, and*

$$(X_I \mid E, X_{\setminus I}) \sim P.$$

Fix any realization of the Poissonized sample size $N \sim \mathrm{Poi}(n)$. Choose $m$ in Lemma H.1. Since for any $\sigma$-smooth $\mathcal{D}_t$, it holds that

$$\frac{\mathcal{D}_t(s)}{\mathcal{U}(\mathcal{X} \times \{0,1\})(s)} = \frac{\mathcal{D}_t(x)}{\mathcal{U}(\mathcal{X})(x)} \cdot \frac{\mathcal{D}_t(y \mid x)}{\mathcal{U}(\{0,1\})(y)} \leq \frac{2}{\sigma},$$

the premise of Lemma H.1 holds with parameter $\sigma/2$ for $P = \mathcal{D}_t, Q = \mathcal{U}(\mathcal{X} \times \{0,1\})$. Consequently, dividing the self-generated samples $\widetilde{s}_1, \cdots, \widetilde{s}_N$ into $N/m$ groups each of size $m$, and running the procedure in Lemma H.1, we arrive at $N/m$ independent events $E_1, \cdots, E_{N/m}$, each with probability at least $1 - (1-\sigma/2)^m \geq 1 - T^{-2}$. Moreover, conditioned on each $E_j$, we can pick an element $u_j \in \{\widetilde{s}_{(j-1)m+1}, \cdots, \widetilde{s}_{jm}\}$ such that

$$(u_j \mid E_j, \{\widetilde{s}_{(j-1)m+1}, \cdots, \widetilde{s}_{jm}\} \setminus \{u_j\}) \sim \mathcal{D}_t.$$

For notational simplicity we denote the set of unpicked samples $\{\widetilde{s}_{(j-1)m+1}, \cdots, \widetilde{s}_{jm}\} \setminus \{u_j\}$ by $v_j$. As a result, thanks to the mutual independence of different groups and $s_t \sim \mathcal{D}_t$ conditioned on $s_{1:t-1}$ (note that we draw fresh randomness at every round), for $E \triangleq \cap_{j \in [N/m]} E_j$ we have

$$(u_1, \cdots, u_{N/m}, s_t) \mid (E, s_{1:t-1}, v_1, \cdots, v_{N/m}) \overset{\text{iid}}{\sim} \mathcal{D}_t.$$

Let us denote $h_{t+1} = O_{t+1}(\widetilde{s}_1, \cdots, \widetilde{s}_N, s_{1:t-1}, s_t)$ the output of the algorithm at time $t$ when $\widetilde{s}_1, \cdots, \widetilde{s}_N$ denotes the hallucinated data points and $s_{1:t-1}, s_t$ denotes the observed data points. We will use the fact that $O_{t+1}$ is a permutation invariant function. Consequently, for each $j \in [N/m]$ we have

$$\underset{s_t \sim \mathcal{D}_t, R^{(t+1)}}{\mathbb{E}} [\ell(h_{t+1}, s_t) \mid E]$$

$$= \underset{s_t \sim \mathcal{D}_t, \widetilde{s}_1, \cdots, \widetilde{s}_N}{\mathbb{E}} \left[ \ell(O_{t+1}(\widetilde{s}_1, \cdots, \widetilde{s}_N, s_{1:t-1}, s_t), s_t) \mid E \right]$$

$$= \underset{v, s_{1:t-1} \mid E}{\mathbb{E}} \left( \underset{s_t, u_1, \cdots, u_{N/m}}{\mathbb{E}} \left[ \ell(O_{t+1}(s_{1:t-1}, v, u_1, \cdots, u_{N/m}, s_t), s_t) \mid E, s_{1:t-1}, v \right] \right)$$

$$= \underset{v, s_{1:t-1} \mid E}{\mathbb{E}} \left( \underset{s_t, u_1, \cdots, u_{N/m}}{\mathbb{E}} \left[ \ell(O_{t+1}(s_{1:t-1}, v, u_1, \cdots, u_{j-1}, s_t, u_{j+1}, \cdots, u_{N/m}, u_j), u_j) \mid E, s_{1:t-1}, v \right] \right) \quad (18)$$

$$= \underset{v, s_{1:t-1} \mid E}{\mathbb{E}} \left( \underset{s_t, u_1, \cdots, u_{N/m}}{\mathbb{E}} \left[ \ell(O_{t+1}(s_{1:t-1}, v, u_1, \cdots, u_{N/m}, s_t), u_j) \mid E, s_{1:t-1}, v \right] \right) \quad (19)$$

$$= \underset{s_t \sim \mathcal{D}_t, R^{(t+1)}}{\mathbb{E}} [\ell(h_{t+1}, u_j) \mid E],$$

where (18) follows from the conditional iid (and thus exchangeable) property of $(u_1, \cdots, u_{N/m}, s_t)$ after the conditioning, and (19) is due to the invariance of the $O_{t+1}$ after any permutation of the inputs. On the other hand, if $s_t', u_1', \cdots, u_{N/m}'$ are independent copies of $s_t \sim \mathcal{D}_t$, by independence it is clear that

$$\underset{s_t, s_t' \sim \mathcal{D}_t, R^{(t+1)}}{\mathbb{E}} [\ell(h_{t+1}, s_t') \mid E] = \underset{s_t, s_t' \sim \mathcal{D}_t, R^{(t+1)}}{\mathbb{E}} [\ell(h_{t+1}, u_j') \mid E], \quad \forall j \in [N/m].$$

Consequently, using the shorthand $u_0 = s_t, u_0' = s_t'$, we have

$$\underset{s_t, s_t' \sim \mathcal{D}_t, R^{(t+1)}}{\mathbb{E}} [\ell(h_{t+1}, s_t') - \ell(h_{t+1}, s_t) \mid E]$$

$$= \frac{1}{N/m + 1} \underset{s_t, s_t' \sim \mathcal{D}_t, R^{(t+1)}}{\mathbb{E}} \left[ \sum_{j=0}^{N/m} (\ell(h_{t+1}, u_j') - \ell(h_{t+1}, u_j)) \mid E \right]$$

$$\leq \frac{1}{N/m + 1} \underset{u_0, \cdots, u_{N/m}, u_0', \cdots, u_{N/m}' \sim \mathcal{D}_t}{\mathbb{E}} \left[ \sup_{h \in \mathcal{F}_\alpha} \sum_{j=0}^{N/m} (\ell(h, u_j') - \ell(h, u_j)) \right]$$

$$\leq \frac{2\alpha}{N/m + 1} \underset{u_0, \cdots, u_{N/m} \sim \mathcal{D}_t}{\mathbb{E}} \underset{\epsilon_1 \cdots \epsilon_{N/m}}{\mathbb{E}} \left[ \sup_{h \in \mathcal{F}_\alpha} \sum_{j=0}^{N/m} \epsilon_j h(u_j) \right]$$

$$\leq \frac{1}{\alpha} \text{Rad}\left( \mathcal{F}_\alpha, N/m \right).$$

The last inequality uses the fact that the algorithm always outputs a function in $\mathcal{F}_\alpha$. Further, we have used the Ledoux-Talagrand contraction inequality.

**Theorem H.2** (Ledoux-Talagrand Contraction). *Let $g : \mathbb{R} \to \mathbb{R}$ be a L-Lipschitz function. For a function class $\mathcal{F}$, denote by $g \circ \mathcal{F}$ the compositions of function in $\mathcal{F}$ with $g$. Then, for all $n$,*

$$\text{Rad}\left( g \circ \mathcal{F}, n \right) \leq L \cdot \text{Rad}\left( \mathcal{F}, n \right).$$

Last inequality follows from the fact that the derivative of the log loss is bounded by $1/\alpha$ when truncated at level $\alpha$. Note that the union bound gives

$$\Pr[E^c] \leq \sum_{j=1}^{N/m} \Pr[E_j^c] \leq \frac{N(1-\sigma)^m}{m}.$$

Thus, the law of total expectation gives

$$\underset{s_t, s_t' \sim \mathcal{D}_t, R^{(t+1)}}{\mathbb{E}} [\ell(h_{t+1}, s_t') - \ell(h_{t+1}, s_t)]$$

$$\leq \underset{s_t, s_t' \sim \mathcal{D}_t, R^{(t+1)}}{\mathbb{E}} [\ell(h_{t+1}, s_t') - \ell(h_{t+1}, s_t) \mid E] + \Pr[E^c] \log(1/\alpha)$$

$$\leq \frac{1}{\alpha} \mathrm{Rad}\left(\mathcal{F}_\alpha, N/m\right) + \frac{N(1-\sigma)^m \log\left(1/\alpha\right)}{m}.$$

The last equation follows from the fact that the output of the algorithm has loss always bounded by $\log\left(1/\alpha\right)$.

We get the desired result by taking the expectation of $N \sim \mathrm{Poi}(n)$, and using $\Pr[N > n/2] \geq 1 - e^{-n/8}$ in the above inequality completes the proof.

# I    Bound on the Perturbation Term

**Lemma I.1** (Perturbation).

$$\mathbb{E}\left[\sum_{i=1}^{N} L(\hat{h}, \tilde{s}_t) - L(h^*, \tilde{s}_t)\right] \leq n \log \alpha$$

*Proof.* Note from the truncation step in Algorithm 1, we have that $L(\hat{h}, \tilde{s}_t) \leq \log\left(\alpha\right)$. We get the desired bound by taking expectations. $\square$

