# OpenReview forum: "Smoothed Analysis of Sequential Probability Assignment"
_NeurIPS.cc/2023/Conference — NeurIPS 2023 spotlight_

### Official Review · Reviewer_4UMy · 2023-06-07

**Soundness:** 3 good
**Presentation:** 2 fair
**Contribution:** 3 good
**Rating:** 6
**Confidence:** 4

**Summary:**

This paper focuses on the contextual sequential probability assignments, and specifically examines cases where the contexts $x_{1:T}$ are generated by $\sigma$-smooth adversaries as introduced in [Haghtalab et al. 2021], and where the labels $y_{1:T}$ are adversarially generated. The primary findings pertain to the minimax regrets under the logarithmic loss, as compared to a generic set of experts $\mathcal{F}$. More specifically, the paper demonstrates the following:

1. If the class $\mathcal{F}$ possesses a scale-sensitive VC-dimension of order $\epsilon^{-p}$, then the information-theoretical minimax regret grows as $T^{p/(p+1)}\cdot \text{poly}\log(T/\sigma)$, which is tight up to polylogarithmic factors.

2. An efficient algorithm exists, given an MLE oracle, that attains regret of order $T^{2/(2+w)}\sqrt{1/\sigma}$, provided the Rademacher complexity of $\mathcal{F}$ is of order $T^{-w}$. For the VC-class, specifically where $w=1/2$, this algorithm yields a regret bound of order $T^{4/5}$.

3. For the VC-class, it demonstrates that the information-theoretical regret is of the order $d\log(T/\sigma)$ up to a constant factor (the same result was also found in concurrent work [Wu et al. 2023]).

The principal contribution of this paper that distinguishes it from previous work on contextual sequential probability assignments is the introduction of an oracle-efficient algorithm that achieves sublinear regret. Although the primary techniques are rooted in the FTPL (Follow The Perturbed Leader) framework as outlined in [Haghtalab et al. 2022], this unique contribution sets a new precedent in this line of research.

**Strengths:**

The primary strength of this work lies in the oracle-efficient algorithm, which presents the first known instance of computationally efficient sublinear regrets under log-loss for general non-parametric classes, albeit using an MLE oracle. This is likely to inspire further research in this field.

**Weaknesses:**

The main weakness is the novelty of the techniques used in this paper. From a purely technical standpoint, Algorithm 1 and its analysis seem to merely mimic those used in [Haghtalab et al. 2022]. In fact, there are many places in the proof that appear to be verbatim copied!

There are also some presentation and technical issues, which I outline below:
- There are many places where the index of summation is $i$ but the summand is written as $s_t$, for instance, in Lemma 4.2. I suggest the authors conduct a thorough proofreading, including the appendix, to correct issues like this;
- I recommend that the authors include a proof for Corollary 4.1.1. It's unclear to me how Theorem 4.1 implies this bound for certain values of $n$, $\alpha$, and $m$;
- The authors claim that a $T^{3/4}$ bound could be achieved for an alternative oracle. This is not clear to me at all. Are you refering to the Theorem 7 of [Block et al., 2022]? If so, it should be properly cited and clearly explained by what you mean "mixed objective function";
-  In line 636 (appendix), it appears that Lemma 3.3 from [Haghtalab et al., 2021] requires $\epsilon> \frac{\sigma}{T\log T}$? The choice of $\epsilon=\frac{\sigma}{T^2}$ does not align with the condition stated in that Lemma. I believe a stronger concentration result, such as Lemma 21 from [Wu et al., 2023], would be required in this context;
- In the proof of Lemma 4.4, presented in Appendix G, you assert that the distributions $P, Q$ are solely dependent on the ensembles $\\{n_0(x),n_1(x)\\}$. However, I contend that this is not a self-evident fact. It is essential, at the very least, to posit that your OPT oracle is permutation invariant with respect to the input sample sequence. This assumption, it should be noted, does not necessarily hold for all optimization methods; stochastic gradient descent, for example, may not exhibit this property.

These are the primary reasons I give a "weak accept".



**Questions:**

I have the following questions for the authors:

1. It seems that your FTPL algorithm relies on an exact ERM oracle. I'm curious, how would the performance or operation of the algorithm change if the oracle was only approximately optimal?
2. Is there any computational lower bounds as in [Haghtalab et al. 2022] that can be proved for the log-loss?
3. Your bound in Theorem 4.1 seems to rely solely on the Rademacher complexity. Does this suggest that your analysis isn't capable of achieving a $\log(T)$ dependency for the VC-class? For instance, consider the linear function $|\langle w,x\rangle|$ with $w,x$ in a unit ball (of a Hilbert space). This class admits a $1/\sqrt{T}$ Rademacher complexity, but it's known that the information theoretical lower bound is $T^{2/3}$.
4. Is it possible to achieve better dependency on $T$ with your FTPL algorithm if the contexts are generated from an i.i.d. process (i.e., with $\sigma=1$)?

**Limitations:**

No issue with negative societal impact.

---

> ### Author Rebuttal · Authors · 2023-08-10
>
> We thank the reviewer for the thoughtful review and the careful reading of our paper. We will fix the typographical errors and omissions.
>
> **Choice of parameters**: Thanks a lot for noticing this. Upon further inspection our proof gives the desired bound with the choice $\epsilon = \sigma / T \log T$ as would work with the original lemma in Haghtalab et al. We will fix this in the revision.
>
> **Permutation Invariance**: We agree that the claim made in the paper would need permutation invariance of the oracle. But in our notation we consider the input to the oracle as the unordered set of historical point and the objective (the average loss) is permutation invariant. In particular, we can achieve the desired permutation invariance by randomly shuffling the data. But we agree that explicitly mentioning this would be helpful and we will elaborate on this in a revision.
>
> **Approximate ERM**: The question regarding the approximate oracle is a good one. Previous works on oracle efficiency have studied the effect of approximation on FTPL type algorithms (See section 6 https://arxiv.org/pdf/1611.01688.pdf). Incorporating (additive) approximations is usually simple but multiplicative approximations are considered challenging and have received study even in the binary case. Even there we don’t have a fully satisfactory solution beyond the linear case. (See for example https://arxiv.org/abs/1804.07837, https://arxiv.org/abs/2102.11050). Studying the appropriate notion of approximate regret in this setting is an interesting avenue for further research.
>
> **Dependence on Rademacher Complexity**: As the reviewer notes the dependence on the Rademacher complexity is a reason for a lack of sharpness in the oracle efficient setting. Unfortunately, this dependence (or on a similar quantity) seems fundamental to FTPL/stability type analysis due to the requirement to analyze the “out of sample generalization” (in our setting, this comes up in analyzing the error between the hallucinated sample and hypothetical coupled random variables). We believe circumventing would require novel ideas and is an exciting avenue of research.
>
> **Computational Lower Bounds**: The main challenge in proving computational lower bounds in the log-loss setting is that most natural algorithms here are fundamentally improper. In general, we do not have good tools to prove lower bounds for improper algorithms (even in the worst-case binary setting). Since the lower bounds in Haghtalab et al are primarily for proper algorithms, they do not port naturally to this setting. Either proving improper algorithm lower bounds or improving oracle efficient regret for improper algorithms is an excellent area for research, whose relevance is further accentuated by oracle efficiency for the log loss introduced in our work.
>
> **IID contexts**: When both the contexts and labels are generated in an iid fashion, the analysis of our regret of our algorithm reduces to the classical analysis of the error of MLE. When the contexts are iid and labels are adversarial, the bound is not immediately clear. Similar setups are studied in online learning (for example in contextual bandits https://arxiv.org/pdf/1606.00313.pdf, https://arxiv.org/pdf/2003.12699.pdf). But usually even in this setting, the analysis is fairly involved. But, the iid context setting is a natural first place towards improved regret bounds.

---

> > ### Comment · Reviewer_4UMy · 2023-08-11
> >
> > I thank the authors for addressing my questions, I have no additional questions.

---

### Official Review · Reviewer_9j8s · 2023-06-19

**Soundness:** 3 good
**Presentation:** 3 good
**Contribution:** 3 good
**Rating:** 6
**Confidence:** 2

**Summary:**

The paper studies the sequential probability assignment problem with a smoothed adversary. The learner sequentially assigns a probability given contexts, which the adversary generates from a distribution that can be far from a known base distribution by a factor of $1/\sigma$. For the problem with the i.i.d. setting, there is an $O(\log T)$ regret method, while we must incur $\Omega(T)$ regret in the pure adversarial setting. Thus, it is well motivated to study what regret is possible in the smoothed setting with respect to $T$ and $1/\sigma$.

The authors provide a general regret bound via reduction to transductive learning. E.g., if the class of functions that map contexts to a probability has the VC dimension of $d$, the regret is $O(d \log (T/\sigma))$, achieving the logarithmic dependence on $T$ and $1/\sigma$. The authors also present an FTPL-style algorithm that is efficient in terms of calls to MLE oracles. It is shown that the algorithm attains $T^{4/5}\sqrt{d/\sigma}$ regret for the VC class.

**Strengths:**

1. The paper studies an interesting online learning problem under the smoothed setting, which is well motivated by the known results in stochastic and adversarial settings.
2. The results imply some interesting consequences. Although I'm not an expert, I found Section 3.2.2 to be interesting, where the authors discuss the benefit of their covering-argument-free idea.

**Weaknesses:**

1. Theorem 3.1, the reduction to transductive learning, seems to be relying heavily on the coupling lemma by Haghtalab et al. [2021].
2. The analysis of the oracle-efficient algorithm in Section 4.1 also appears to be relying on Haghtalab et al. [2022].

**Questions:**

1. I would appreciate it if the authors could describe intuitive technical differences from Haghtalab et al. [2021, 2022].
2. The scale-sensitive VC dimension appears to be related to the fat-shattering dimension. Is there some connection between them?

#### Minor comments
- In Algorithm 1's input, capitalization is inconsistent.
- In Algorithm 1, line 2, a period or a line break is missing before "Call the oracle..."
- I prefer references arranged in alphabetical order.
- Hyperlinks do not work.

**Limitations:**

Although limitations are not explicitly discussed, the range to which their results apply seems clear from the description.

---

> ### Author Rebuttal · Authors · 2023-08-10
>
> We thank the reviewer for the thoughtful comments. We will incorporate corrections to the typographical errors in a revision.
>
> **Differences between Haghtalab et al 21,22 and our work**: The main focus of the work by Haghtalab et al 21, 22 was on the case of the binary loss. Haghtalab et al 21 focused on the case of the binary loss and VC classes and presented a statistical rate for the smoothed setting matching the rates for the iid case. In our work, we focus on the log loss which does not fall into their framework because of the poor Lipschitz constant and unboundedness. Furthermore, their original result can be seen as a covering based argument while our general result is a direct reduction from the smoothed setting to the transductive setting. Our work goes beyond finite covering numbers. This allows our work to extend to non-parametric settings and others where the covering numbers do not capture the regret in the  log-loss setting. To get these stronger results, our approach differs entirely from Haghtalab et al. Rather than looking at coverings, our main statistical result establishes a reduction from the smoothed online setting to a version of transductive setting. In particular, we show that the regret is small as long as the notion of transductive regret is small without the need to restrict to settings with small covering numbers.
>
> Haghtalab et al 22 focuses on the oracle efficiency (also for the binary loss or finite regression loss). The main algorithmic difference between our result and theirs is the choice of the oracle. In our paper, we focus on the MLE oracle which just outputs the best model in the class on the data while their paper uses a stronger notion of what is commonly referred to as a mixed binary-regression oracle. The difference is important in the case of the log loss because the MLE oracle can be seen as a natural subroutine in statistical analysis while the general mixed binary-regression oracle is unnatural in most settings including the log loss setting. Technical challenges arise from the fact that the log loss is unbounded and non-Lipschitz which was a necessary requirement from previous work. Furthermore, a technical challenge in handling the MLE oracle corresponds to a different way of controlling the perturbation term in the regret.
>
> **On the use of coupling lemma**: Indeed, the coupling lemma of Haghtalab et al ‘21 was a fundamental tool in the analysis of smoothed online learning. Since then, every work in this space published at top tier conferences such as NeurIPS, STOC/FOCS, and COLT (which constitutes 10s of papers) has relied on that coupling lemma. This should be taken as the evidence of the unusual versatility of the coupling lemma of Haghtalab et al ’21, but not to underestimate contributions of a long line of work on smoothed analysis since Haghtalab et al ’21.
>
> Furthermore, the original result of Haghtalab et al  ‘21 can be seen as a covering based argument while our general result is a direct reduction from the smoothed setting to the transductive setting. Our work goes beyond finite covering numbers. This allows our work to extend to non-parametric settings and others where the covering numbers do not capture the regret in the  log-loss setting. To get these stronger results, our approach differs entirely from previous work. Rather than looking at coverings, our main statistical result establishes a reduction from the smoothed online setting to a version of transductive setting. In particular, we show that the regret is small as long as the notion of transductive regret is small without the need to restrict to settings with small covering numbers. Our reduction to the transductive setting is a novel and promising approach that we expect will have an equally versatile role to play (compared to the coupling lemma of Haghtalab et al ‘21) for the study of statistical and computational perspective of unbounded losses and losses with curvature.
>
>
> **Scale-sensitive Dimension and Fat shattering dimension**: The notions of complexity are different terminology arising from different communities. For example, compare definition 3.2 in https://home.ttic.edu/~tewari/lectures/lecture15.pdf  and section 2.4 in https://arxiv.org/pdf/2202.04690.pdf

---

> > ### Comment · Reviewer_9j8s · 2023-08-16
> >
> > I appreciate the authors' thorough response. My concerns have been adequately addressed. While my expertise in this field is limited, I believe that this paper is technically solid and deserves acceptance.

---

### Official Review · Reviewer_X6XT · 2023-07-05

**Soundness:** 3 good
**Presentation:** 3 good
**Contribution:** 3 good
**Rating:** 6
**Confidence:** 4

**Summary:**

The paper considers the sequential probability assignment problem, which is the following:

The algorithm (forecaster), based on past context and outcomes, must assign probabilities to 0-1 values for the next outcome given the latest context. The forecaster competes against a reference class of predictor functions (experts) and aims to suffer low regret compared to the best of these functions. For a logarithmic loss incurred for incorrect predictions, regret is defined as the difference in the total log-loss of the forecaster and the best of the experts.

When the contexts are drawn independently from an unknown distribution, and the class of predictor functions is special (say, has a bounded VC dimension), the sequential probability assignment has low (i.e. sublinear) regret. However, if the contexts are adversarial, there are impossibility results even for simple classes of predictors.

Therefore, following Haghtalab et al. (2021), the authors study a smoothed version of the problem where contexts can only be chosen adaptively from the set of sigma-smoothed distributions. They study how the properties of the class of the predictor functions like bounded “scale-sensitive” VC dimensions affect the regret.

The main tool used is a powerful coupling lemma from Haghtalab et al. (2021) that converts the problem with an adaptive sequence of t contexts to a problem with t*K uniformly and independently distributed contexts (the length of the sequence increases but they are no more adaptive). Thus, results for i.i.d. samples may be applied where the regret is essentially logarithmic in T (number of time steps), which is the first result of the paper.

The second question studied is an algorithmic approach to solving the problem using calls to MLE. A follow the perturbed leader type algorithm using ideas generalized from Haghtalab et al. (2022) is used. Here, however, the regret is not logarithmic but T^0.8. This is the second main result in this paper.

**Strengths:**

The paper is well-written and the considered problem is natural. The results are interesting.

**Weaknesses:**

It seems the first main result of this paper is already presented in a paper that has been accepted at COLT 2023 (Wu et al.) Even though this seems to be independent work of a different group of authors, I would still find it strange if the same result was now also accepted at NeurIPS, in particular because the proofs are quite similar. Both rely essentially on the coupling arguments from the work of Haghtalab et al. I also have to say that my impression is that the known coupling argument is really the key ingredient and the proof in this submission is "only" an adaption of the arguments by Haghtalab to the log-loss setting. Maybe the authors could point out more clearly if there were significant challenges in this adaption.

The second main result is not contained in the COLT paper. However, it unfortunately only shows a regret of T^0.8 and no logarithmic one and it also follows known algorithmic approaches from the literature like follow the perturbed leader.

**Questions:**

--

**Limitations:**

n.a.

---

> ### Author Rebuttal · Authors · 2023-08-10
>
> We thank the reviewer for their review.
>
> **Regarding the concurrent work by Wu et al**: As the reviewer pointed out that both papers are concurrent and independent works. We agree that at first sight the ideas in the statistical aspect of our paper are related to the ones in Wu et al 23 and are both inspired by the coupling lemma from Haghtalab et al 21. As we explain below, the overlap between the results are minor and limited to one of our statistical rates, our statistical rates are stronger than Wu et al in general, and one of our primary contributions (computational perspective) forms a very novel setting that has not been studied before. In short, our work has independent insights, results, contributions, and open directions than Wu et al. 23 and will draw independent readership and interest from the field.
>
> The small overlap between the results is only limited to the rates for the case of VC classes (Corollary 3.4.1). Even in the statistical case more broadly, our work goes beyond VC classes as opposed to Wu et al, who only considered the case of finite covering numbers. This allows our work to extend to non-parametric settings where the covering numbers do not capture the regret. To get these stronger results, our approach differs entirely from Wu et al. Our main statistical result establishes a reduction from the smoothed setting to a transductive setting. Interesting consequences of this approach are discussed in Section 3.2.2.
>
> **Algorithmic Challenges**: A major contribution of our work is the study of oracle-efficient online learning for the log-loss, which has not been considered by Wu et al. Indeed, oracle efficiency has not been considered at all for the log loss even for worst-case settings. Considering oracle-efficiency for log-loss is an important line of research as the ERM oracle corresponds to the maximum likelihood estimator (MLE) that is commonly used in practice.  Our paper presents sublinear regret oracle efficient algorithms (which does not have an analog in Wu et al 2023). ​​In our opinion, this is a significant contribution.
>
> We acknowledge the reviewer’s concern regarding regret rates. As we state in our section 5, we believe that obtaining subpolynomial bounds is an interesting direction for future work. We also provide discussions and some evidence that highlights why any MLE oracle (and FTPL analysis) will face significant technical challenges in obtaining subpolynomial regret bounds, due to these methods’ reliance on $\sqrt{T}$ “out of sample generalization'' variance term.  We believe that this discussion and our proof approach will be insightful more broadly for online algorithm design. Proving lower bounds is also challenging, as log-loss necessitates considering improper algorithms which are historically challenging for obtaining lower bounds. This puts the problem in an intriguing situation that warrants further investigation. We believe that significant progress requires perspectives and the publication of our work will bring this to the attention of the community.
>
> **On the algorithmic framework of FTPL**:  One of our main technical contributions is an oracle-efficient algorithm that taps into the MLE oracle to achieve sublinear regret for the log-loss. We emphasize that most work on oracle efficiency considers a “mixed loss” oracle that minimizes a signed combination of the historical losses. This is unnatural in the log loss setting and a major technical consideration that sets our work apart from other work on oracle-efficient online learning is to work just with MLE oracles due to the natural connection to statistical estimation.
>
> Further we believe "known algorithmic approaches from the literature like follow-the-perturbed leader" again minimizes our contributions. In particular, algorithms in the oracle efficient setting are forced be similar to FTPL, since the algorithm is only promised a "leader" oracle. The fact that all oracle-efficient algorithms utilize the FTPL framework is a testament to its fundamental nature. The main challenge in designing oracle efficient algorithms is getting regret and running times roughly logarithmic in the number of experts (note that the basic FTPL analysis gives polynomial dependence on the number of experts). Since this (provably) cannot be done in generality each paper adapts the FTPL framework to the particular setting.
>
> **On the use of coupling lemma**: Indeed, the coupling lemma of Haghtalab et al ‘21 was a fundamental tool in the analysis of smoothed online learning. Since then, every work in this space published at top tier conferences such as NeurIPS, and COLT (which constitutes 10s of papers) has relied on that coupling lemma. This should be taken as the evidence of the unusual versatility of the coupling lemma, but not to underestimate contributions of a long line of work on smoothed analysis. Additionally, see the comments above regarding the reduction between smoothed analysis and transductive learning for further differences between our work and prior work.
>
> **On treatment of concurrent literature**: As demonstrated above, our results are novel and have only minor overlap with Wu et al. Indeed, the work of Wu et al did not consider  computational aspects to smoothed analysis. Even for statistical analysis, our approach presents a more general approach that extends to general losses. In short, our work has independent insights, results, contributions, and open directions than Wu et al. 2023 and will draw independent readership and interest from the field.
>
> The publication of the concurrent works (with a minor overlap) in other conferences is only evidence that the community is very interested in this line of work. The first versions of our work were made public within days of each other. We believe that rejecting a work for a minor overlap with concurrent work is a deviation from standard practice in the review process and would be counterproductive for the research community.

---

> > ### Comment · Reviewer_X6XT · 2023-08-16
> >
> > Thank you for the clarification. I understand now that the overlap with the paper by Wu et al. is not as large as I had originally thought. Hence, I have raised my score.

---

### Official Review · Reviewer_oQa8 · 2023-07-08

**Soundness:** 4 excellent
**Presentation:** 4 excellent
**Contribution:** 3 good
**Rating:** 7
**Confidence:** 3

**Summary:**

The paper discusses smoothed analysis of probability assignments in an online setting. The paper shows how the problem can be reduced to a transductive setting and obtains an upper bound on the regret using covering numbers which is further bounded by the scale sensitive VC dimension. The results are instantiated for various classes of functions, including VC function class and the non-parametric class where the scale sensitive VC dimension grows as \epsilon^{-p} at scale \epsilon.

On the algorithmic side, the authors propose an oracle efficient FTPL style algorithm to predict the probability assignments and show weaker regret bounds for the same (by bounding the Rademacher complexity)

**Strengths:**

- The paper is very well written and is very clear and easy to follow.
- The paper is the first to consider Oracle efficient algorithms for the sequential prob. assignment problem
- The paper creatively combines earlier ideas (Coupling Lemma) to reduce the smoothed adversary setting to a transductive setting.


**Weaknesses:**

Please see questions below

**Questions:**

- The authors can elaborate the results of Wu.et al [2023] and highlight the differences and similarities (both in terms of results and techniques).
-  Is Algorithm 1 actually implementable in practice - especially without the knowledge of \sigma, the smoothness parameter?
- How do the results change in case the adversary is realizable w.r.t to the class F that the regret is measured against?
- How critical is the assumption on uniform distribution over X. For instance if X does not support a uniform distribution, how does Algorithm 1 and it's guarantees change?

---

> ### Author Rebuttal · Authors · 2023-08-10
>
> We thank the reviewer for the thoughtful comments.
>
> **Regarding the concurrent work by Wu et al**: As the reviewer pointed out that both the papers were concurrent and independent works. We agree that the ideas in the statistical aspect of our paper are related to the ones in Wu et al 2023 and are both inspired by the coupling lemma from Haghtalab et al 2021.
>
> The small overlap between the results of these two papers is only limited to the statistical rates for the case of VC classes (the statement of Corollary 3.4.1). Even in the statistical case, our work goes beyond VC classes as opposed to Wu et al, who only considered the case of finite covering numbers. This allows our work to extend to non-parametric settings and others where the covering numbers do not capture the regret in the  log-loss setting. To get these stronger results, our approach differs entirely from Wu et al. Rather than looking at coverings, our main statistical result establishes a reduction from the smoothed online setting to a version of transductive setting. In particular, we show that the regret is small as long as the notion of transductive regret is small without the need to restrict to settings with small covering numbers. Interesting consequences of this covering-number free approach are discussed in Section 3.2.2.
>
>
> A major contribution of our work is the study of oracle-efficient online learning for the log-loss, which has not been considered by Wu et al. Indeed, to the best of knowledge, oracle efficiency has not been considered at all for losses such as the log loss even for worst-case settings. Considering oracle-efficiency for log-loss is an important and natural line of research as the ERM oracle corresponds to the maximum likelihood estimator (MLE) that is commonly used in practice.  Our paper presents sublinear regret oracle efficient algorithms for the log loss (which does not have an analog in Wu et al 2023).  ​​In our opinion, the algorithmic questions in our paper are a significant contribution.
>
> **Implementation in practice and Smoothness parameter**:
> We believe that the algorithm is very reasonable to implement in practice when given access to a class of models that one can optimize over. Note that the algorithm just choses some samples from the base distribution and adds it to the training set and uses the optimization oracle. For many classes of interest such as neural networks one can implement optimization oracles using (stochastic) gradient descent (in practice).
>
> Further, we clarify that the exact knowledge of $\sigma$  is not needed by our approach. Our algorithms and regret bounds can work with any approximation of sigma value that is a lower bound of the real up to constant multiplicative factors. This corresponds to settings where the world is more smooth than we give it credit. Even when we have extremely poor upper and lower bounds, we can use hedging to still get non-trivial regret with only a minor blow up in computation. We will provide more details next as to how we work with knowledge of approximate sigma.
>
> In general, given (loose) upper and lower bounds on the exact value, we can use a geometric doubling approach to deal with the unknown $\sigma$. To be specific, one could construct experts, where each expert runs a local version of our algorithm with parameters $ \sigma_i = \sigma_{low} 2^i$. Here $\sigma_{low}$ is a loose lower bound on $\sigma$. We maintain experts till $\sigma_i = \sigma_{high}$ which is a loose upper bound on $\sigma$. We then run Hedge on these experts. Note that the parameter of the best expert satisfies $\sigma/2 \leq   \sigma_i \leq 2 \sigma$ , so its regret matches the regret of the same algorithm that runs on true $\sigma$ up to a constant factor. Therefore, the expected regret of this meta algorithm is comparable to the bound in with known $\sigma$, with an additive term of order at most $\sqrt{T \log \log (\sigma_{high} / \sigma_{low} )}$. This could potentially be improved using a more aggressive step size for the Hedge meta algorithm.
>
> **Realizability**: For the statistical rates, realizability (appropriately defined) does not affect the rates significantly.But for the oracle efficient case this is an interesting question. Even for the binary loss case, this is not fully understood in the sense that we don't know an algorithm that achieves oracle efficient fast rates in the realizable case. Understanding this in the log loss case is an excellent avenue for improving the rates.
>
> **Uniformity on $\mathcal{X}$**: The notion of smoothness with respect to an arbitrary measure in general without the domain to have a uniform distribution. The algorithm 1 generalizes in a natural way to this setting by sampling from the base measure $\mu$ instead of the uniform distribution. The analysis also remains the same (by and large) but some care needs to taken in order to define appropriate notions (such as Poisson processes) that we use in the analysis to the arbitrary domain.

---

> > ### Comment · Reviewer_oQa8 · 2023-08-20
> >
> > Thanks for the author's response. I have read the rebuttal.  The clarifications on certain points in the rebuttal such as the practical implementation, arbitrary domain, etc can be added to the paper/supplementary as appropriate.

---

> ### Comment · Area_Chair_KTBK · 2023-08-18
>
> Dear Reviewer oQa8,
>
> Could you please acknowledge the author's rebuttal?
>
> Thank you,
> Your AC

---

### Official Review · Reviewer_w3DK · 2023-07-25

**Soundness:** 4 excellent
**Presentation:** 3 good
**Contribution:** 3 good
**Rating:** 7
**Confidence:** 3

**Summary:**

This paper studies the problem of sequential probability assignment in the smoothed setting. In particular, a learner receives labelled examples sequentially, where the contexts are drawn from some smooth distribution which can otherwise be chosen adversarially (in each step) and the labels can be chosen adversarially. The goal of the learner is to minimize the regret (minimax rate of excess error with respect to the optimum hypothesis on the subsequence seen so far).

The first main result provided is that the optimum regret for the problem is characterized by the optimum regret of a different problem where the adversary chooses a priori the set of contexts that they will have to choose from throughout the interaction with the learner. Using the demonstrated relationship between the two problems, the authors provide an upper bound on the regret in the smoothed setting, which involves the scale sensitive VC-dimension of the hypothesis class and a number of free parameters that can be chosen accordingly to adapt the bound to different regimes for the VC-dimension. For VC-classes the provided adapted bound is essentially tight.

The second main result provided is an oracle-efficient algorithm achieving sublinear regret on VC-classes (and more generally on classes with polynomially decaying Rademacher complexity), provided access to an ERM oracle for the hypothesis class considered. The algorithm follows the approach of Follow-the-Perturbed-Leader, which takes advantage of the trade-off between the stability of the algorithm's intermediate states and the excess error due to perturbing the current set of samples (by adding a number of hallucinated uniformly random data points).

**Strengths:**

The paper demonstrates that the smoothed analysis framework is relevant to providing provable guarantees for the problem of sequential probability assignment, hence initiating (and motivating) the (further) study of the problem through the lens of smoothed analysis. The results provided include essentially tight bounds on the (smoothed) regret for VC-classes and two main conceptual contributions: First, characterizing the minimax regret for the problem considered in terms of the minimax regret of another relevant problem (rather than some combinatorial notion of dimension, which might even be hopeless in general). Second, repurposing ERM as an oracle to be exploited by algorithms with sublinear regret.

Overall, the results are presented with clarity and sufficient detail, and provide bounds for the considered problem that are fairly general and adaptable to hypothesis classes with different properties.

**Weaknesses:**

The first weakness of the paper is that the oracle-efficient algorithm proposed is not necessarily efficient, due to the potentially high complexity of the ERM oracle, especially since the labels are chosen adversarially. In other words, it is not clear whether assuming oracle access to an ERM is reasonable. A discussion on existing or simple positive results regarding the oracle's implementation or even a pointer to empirical results that demonstrate its success on a relevant setting would be appreciated.

Furthermore, many of the technical contributions of the paper are not discussed in the main text. For example, the only hint provided for the proof of Theorem 3.1 in the main text is that it uses Theorem 2.1. In particular, it is not clear what (if any) technical obstacles arise when one tries to apply Thm 2.1 to this setting, or if adapting some approach from prior work is sufficient. Similarly, a proof sketch for Theorem 3.2 would be helpful for the reader.

Overall, while the problem considered is well-motivated and the results are concrete, it is not clear whether there are strong conceptual and technical contributions relatively to prior work (smoothed analysis has already been applied to online learning and most of the tools used to demonstrate the results existed in the literature, at least in some less general form -- e.g., Theorem 2.1 and the idea of decomposing the stability term in Haghtalab et al. [2022]).

-- Most of my main concerns where addressed in the rebuttal and for this reason I increased my score from 6 to 7.

**Questions:**

My main questions are related to the two main points raised in the weaknesses section. Given the concreteness of the provided results, I would be happy to increase my score if the issues I pointed to are sufficiently addressed by the authors' answer.

1. Are there some simple function classes for which the ERM oracle can be efficiently implemented under the marginal assumptions that correspond to the considered setting? Alternatively, is there a(n empirical, conceptual or theoretical) reason for which one might expect to heuristically obtain an approximate ERM oracle?

2. What are the main technical hurdles arising when one tries to instantiate prior techniques in the considered setting? In which ways does the technical work differ from prior work?

I have also found a small number of typos:
- line 294: likliehood $\to$ likelihood
- lines 306-307: i=1 $\to$ t=1 (3 times, one for each summation)
- line 318: at in $\to$ in
- line 321: refers $\to$ refer
- line 344: achieve $\to$ achieved

**Limitations:**

Yes

---

> ### Author Rebuttal · Authors · 2023-08-10
>
> We thank the reviewer for a thoughtful review. We will incorporate the suggested typographical and expository corrections.
>
> **Applicability of oracle-efficiency**: The oracle-efficient framework is important because it allows us to directly tap into existing deployed algorithms, without having to design and implement an algorithm from scratch. These sub-routine algorithms can be heuristics and do not have to be provably efficient. Modern computer science is full of such heuristics that perform exceedingly well in practice even when hardness barriers exist in theory; a great example of this is deep learning. The oracle-efficient method for designing online algorithms has been extremely popular recently and has seen a lot of use in varied contexts such as contextual bandits and reinforcement learning (see https://vowpalwabbit.org/) and is even used in production. We see our work as following this line of work to design online algorithms. In particular, our paper elucidates the relative complexity of maximum likelihood estimation and sequential conditional density estimation.
>
> Note that the oracle in our algorithm is called on either “smoothed” instances given by the adversary, or random instances sampled from the uniform distribution. In such settings,  hardness results usually do not hold since they are proven mostly for worst case instances. Therefore, when implementing our algorithms in practice, instead of using an oracle that is provably efficient for all worst-case inputs, it suffices to have a weaker oracle that performs reasonably well on “average” case instances.
>
> Further, for many practical classes of conditional probability densities, heuristic algorithms (for example bases on deep learning) are often used in practice. One perspective on this line of work is developing machinery to convert algorithms for heuristic optimization into provable algorithms for sequential decision making
>
>
> **On technical hurdles of our work and contributions relative to prior work**:
> Prior work focused on the case of the binary loss and VC classes and presented a statistical rate for the smoothed setting matching the rates for the iid case. In our work, we focus on the log loss which does not fall into their framework because of the poor Lipschitz constant and unboundedness. Furthermore, their original result can be seen as a covering based argument while our general result is a direct reduction from the smoothed setting to the transductive setting. Our work goes beyond finite covering numbers. This allows our work to extend to non-parametric settings and others where the covering numbers do not capture the regret in the  log-loss setting. To get these stronger results, our approach differs entirely from Haghtalab et al. ‘22. Rather than looking at coverings, our main statistical result establishes a reduction from the smoothed online setting to a version of transductive setting. In particular, we show that the regret is small as long as the notion of transductive regret is small without the need to restrict to settings with small covering numbers. Interesting consequences of this covering-number free approach are discussed in Section 3.2.2.
>
>
> The algorithmic questions in our paper require new technical innovations. To the best of knowledge, oracle efficiency has not been considered at all for losses such as the log loss even though we believe this is a natural setting for oracle efficiency since the ERM oracle corresponds to the maximum likelihood estimator (MLE). Technical challenges arise from the fact that the log loss is unbounded and non-Lipschitz which was a necessary requirement from previous work. Furthermore, a main technical contribution is techniques to handle the MLE oracle (which differs from the previous work which uses a more general regression oracle). Technically, this corresponds to a different way of controlling the perturbation term in the regret.

---

> > ### Comment · Reviewer_w3DK · 2023-08-11
> >
> > Thank you for your detailed response. I have increased my score in light of your clarifications.

---

### Decision · Program_Chairs · 2023-09-21

**Decision:**

Accept (spotlight)

**Comment:**

This paper analyses the regret of sequential probability assignment problems with binary labels, when the labels are adversarial but feature distributions are smoothed. Reviewers agree that this is a strong contribution to the recent literature on characterizing such problems by weakening the strong adversarial setting. The most important aspect to address is to explain the significance of the theorems in plain language, to improve their appreciation by the readership. The authors are also advised to more carefully compare with the Haghtalab et al 2022 and the concurrent  Wu et al 2023 papers, particularly in terms of the technical tools used. Discussions with reviewers, e.g., on the practical usefulness of oracle access to ERM and where changing the uniformity assumption would affect the results, should also ideally find their way to the paper. Lastly, the paper's mathematical notation could benefit from a thorough proofreading pass, e.g., to make sure the summation indices are correct.